# Exploiting Open-World Data for Adaptive Continual Learning

## Abstract

Continual learning (CL), which involves learning from sequential tasks without forgetting, is mainly explored in supervised learning settings where all data are labeled. However, high-quality labeled data may not be readily available at a large scale due to high labeling costs, making the application of existing CL methods in real-world scenarios challenging. In this paper, we delve into a more practical facet of CL: open-world continual learning, where the training data comes from the open-world dataset and is partially labeled and non-i.i.d. Building on the insight that task shifts in continual learning can be viewed as transitions from in-distribution (ID) data to out-of-distribution (OOD) data, we propose OpenACL, a method that explicitly leverages unlabeled OOD data to enhance continual learning. Specifically, OpenACL considers novel classes within OOD data as potential classes for upcoming tasks and mines the underlying pattern in unlabeled open-world data to empower the model's adaptability to upcoming tasks. Furthermore, learning from extensive unlabeled data also helps to tackle the issue of catastrophic forgetting. Extensive experiments validate the effectiveness of OpenACL and show the benefit of learning from open-world data.

## 1 Introduction

Continual learning, unlike conventional supervised learning which learns from independent and identically distributed (i.i.d.) data, allows machines to continuously learn a model from a stream of data with incremental class labels. One of the main challenges in CL is to tackle the issue of the *catastrophic forgetting*, i.e., prevent forgetting the old knowledge as the model is learned on new tasks (De Lange et al., 2021). Although many approaches (e.g., methods based on data replay (Rebuffi et al., 2017; Lopez-Paz & Ranzato, 2017), weight regularization (Kirkpatrick et al., 2017; Li & Hoiem, 2017)) have been proposed to tackle catastrophic forgetting in CL, they rely on an assumption that a *complete* set of *labeled* data is available for training and focus on a supervised learning setting. Unfortunately, this assumption may not hold easily in real applications when obtaining high-quality sample-label pairs is difficult, possibly due to high time/labor costs, data privacy concerns, lack of data sources, etc. This is particularly the case for CL where the number of classes increases during the learning process.

To effectively learn CL models from limited labeled data, recent studies (Smith et al., 2021; Wang et al., 2021; Lee et al., 2019) suggest leveraging the semi-supervised learning (SSL) technique for CL to learn from both labeled and unlabeled data. The idea of SSL is to improve model performance by using limited labeled data and a larger amount of unlabeled data. In real applications, obtaining a steady stream of labeled data can be very expensive and time-consuming for CL, especially in new or rapidly evolving domains. However, obtaining large amounts of unlabeled data is relatively easier. SSL has proven effective and is applied to many tasks including CL. Specifically, Wang et al. (2021) considers a typical SSL setting where labeled and unlabeled data are assumed to be i.i.d. so that the unlabeled data can be leveraged to help improve the model performance. However, the i.i.d. assumption is commonly violated as the unlabeled data is usually acquired from different sources and there exist distributional shifts between unlabeled and labeled data. In a worse case, the unlabeled data may be of low quality and contain large proportions of out-of-distribution (OOD) data. To address this, Lee et al. (2019); Smith et al. (2021) extend Wang et al. (2021) to non-i.i.d. settings by considering the existence of OOD data in the dataset. As an example, Smith et al. (2021) proposes a method that learns two models, with one for distinguishing and eliminating OOD data

and the other for predictions. Notably, it treats all seen classes up to the current task as ID data and uses a manually set threshold to reject OOD samples. To maintain high prediction accuracy while preventing forgetting, it then actively identifies unlabeled data that is relevant to the incremental task and repeatedly trains the model using both labeled and identified unlabeled data.

Unlike the existing methods that only leverage ID unlabeled samples to enhance model performance, we argue that unlabeled OOD data can also be useful in CL when there are distribution shifts between tasks. It is based on the observation that the novel classes in OOD data for previous/current tasks may become training classes in future tasks, e.g., an unseen class "car" for the current task could belong to the task classes for upcoming tasks. Instead of identifying and eliminating OOD samples during training, we may leverage them to adapt a model to a new task and improve the model performance in CL. Our paper is based on this idea, where we aim to exploit the patterns of unlabeled data, especially novel classes in OOD data, and use them to adapt the model to future tasks in CL; as opposed to the previous works that simply reject those OOD samples with low confidence.

Specifically, this paper considers open semi-supervised continual learning (Open SSCL). The goal is to learn a model continuously from both labeled and unlabeled data in an open world without forgetting, and meanwhile effectively utilizing unlabeled data to adapt to novel classes. Unlike previous SSCL problems that only use unlabeled data to prevent catastrophic forgetting, unlabeled data in Open SSCL should also be used to adapt the model to novel classes (new tasks) under distribution shifts. In other words, Open SSCL aims to use easy-to-obtain unlabeled open-world data to improve CL model performance on past, current, and future tasks.

Toward this end, we propose an **Open** semi-supervised learning framework **A**dapting the model to new tasks in **C**ontinual **L**earning (OpenACL). We introduce a prototype-based learning method to learn a generalized representation of unlabeled data and adapt the model to a new task while shifting the tasks. Besides, the ID samples in extensive unlabeled data can also be leveraged by the prototypes to mitigate catastrophic forgetting. Our contributions can be summarized as follows:

- We formulate a problem of open semi-supervised continual learning (Open SSCL). It is motivated by the fact that real data in practice mostly contains limited labeled data and large-scale unlabeled data, with the existence of OOD data in unlabeled data. Notably, instead of eliminating OOD data, Open SSCL can utilize OOD samples to enhance model performance on new tasks.

- We propose a method called OpenACL to solve the Open SSCL problem. It maintains multiple prototypes for seen tasks and reserves extra prototypes for unseen tasks. Both labeled and unlabeled data are learned to improve the adaptation ability and tackle forgetting for prototypes.

- We conduct extensive experiments to evaluate OpenACL. We also extend the existing CL methods to the Open SSCL setting and compare them with ours under a fair environment. The online continual learning results show that OpenACL consistently outperforms others in adapting to new tasks and addressing catastrophic forgetting.

## 2 RELATED WORK

This paper is closely related to the literature on continual learning, semi-supervised learning, and open set/world problems. We introduce each topic and discuss the differences with our work below.

**Continual Learning (CL).** The goal is to learn a model continuously from a sequence of tasks (non-stationary data). One of the challenges in CL is to overcome the issue of catastrophic forgetting, i.e., prevent forgetting the old knowledge as the model is learned on new tasks. Various approaches have been proposed to prevent catastrophic forgetting, including regularization-based methods, rehearsal-based methods, parameter isolation-based methods, etc. Specifically, *regularization-based* methods prevent forgetting the old knowledge by regularizing model parameters; examples include Elastic Weight Consolidation (Kirkpatrick et al., 2017), Synaptic Intelligence (Zenke et al., 2017), Incremental Moment Matching (Lee et al., 2017), etc. In contrast, *rehearsal-based* methods (Rebuffi et al., 2017; Lopez-Paz & Ranzato, 2017; Saha et al., 2021) tackle the problem by reusing the old data (stored in a memory-efficient replay buffer) in previous tasks during the training process. Unlike these approaches where a single model is used for all tasks, *parameter isolation-based* methods (Mallya & Lazebnik, 2018) aims to improve the model performance on all tasks by isolating parameters for specific tasks.

Note that all the above methods were studied in the classic supervised learning setting. In contrast, our paper considers an open semi-supervised environment with not only labeled data but also unlabeled data that is possibly OOD.

**Semi-Supervised Learning (SSL).** It aims to learn a model from both labeled and unlabeled data, and the labeled data is usually limited while the unlabeled ones are sufficient. *Pseudo-labeling-based methods*, as discussed by Xie et al. (2020); Xu et al. (2021); Sohn et al. (2020), initially train models using labeled data and subsequently assign virtual labels to the unlabeled data. Then the model with improved performance is learned from those sample-pseudo-label pairs. On the other hand, *consistency regularization-based* methods (Sajjadi et al., 2016; Meel & Vishwakarma, 2021) learn to ensure consistency across different data. They augment the unlabeled data by generating different views of data (e.g., by rotation, scaling, etc.), and a model is then trained on the augmented data via regularized optimization such that the predictions for different views are consistent.

While SSL has shown success in many tasks, its application to CL is less studied. Because unlabeled data in practice may not follow the identical distribution as the labeled data and they may come from different classes, SSL methods introduced above may not perform well in real applications. This paper closes the gap where we focus on CL and extend SSL to the open setting.

**Open-Set & Open-World Recognition.** It considers scenarios where the data observed during model deployment may come from unknown classes that do not exist during training. The goal is to not only accurately classify the seen classes, but also effectively deal with unseen ones, e.g., either distinguish them from the seen classes (open-set problem) or label them into new classes (open-world problem). The existing methods for open-set recognition include traditional machine learning-based methods (Bendale & Boult, 2015; Mendes Júnior et al., 2017; Rudd et al., 2017) and deep learning-based methods (Dhamija et al., 2018; Shih et al., 2019; Yu et al., 2017; Yang et al., 2019). Each can further be classified into discriminative model-based and generative model-based methods, depending on whether the unknown classes are detected by calibrating the classification logistics (Yoshihashi et al., 2019; Rozsa et al., 2017; Hassen & Chan, 2020) or by learning distributions of known classes (Ge et al., 2017; Yu et al., 2017; Neal et al., 2018; Jo et al., 2018; Yang et al., 2019).

In this paper, we consider open-world settings but primarily focus on semi-supervised continual learning, where the model is trained from a sequence of tasks and the training dataset includes both labeled and unlabeled data.

**Open-Set/World Semi-Supervised Learning.** It combines both open-set/world recognition and SSL. The goal is to train a model from both labeled and unlabeled data, where the unlabeled data may contain OOD samples. One of the challenges is to make SSL less vulnerable to OOD samples. To this end, most existing methods (Guo et al., 2020; Saito et al., 2021; Lu et al., 2022) first detect OOD samples, which are then rejected or re-weighted to ensure performance. For example, Guo et al. (2020) proposes a method that selectively uses unlabeled data by assigning weights to unlabeled samples. OpenMatch (Saito et al., 2021) integrates a One-Vs-All detection scheme to filter out OOD samples in SSL training loops. Cao et al. (2022) extends the open-set SSL and proposes open-world SSL, which requires actively discovering novel classes. This work is generalized in (Rizve et al., 2022; Tan et al., 2023) where novel classes are discovered using a pairwise similarity loss.

Our paper extends Open SSL to CL. In particular, we note that the data from untrained tasks in CL can indeed be viewed as OOD samples. Based on this, we study the open semi-supervised continual learning (Open SSCL) problem. We will illustrate how the unlabeled data can be leveraged in Open SSCL to mitigate catastrophic forgetting and adapt a model to new tasks.

## 3 PROBLEM FORMULATION

In this section, we formulate the problem of open semi-supervised continual learning (Open SSCL).

Consider a continual learning problem that aims to learn a model from a sequence of $k$ tasks $T = \{T_1, ...T_k\}$. Let $\mathcal{D} = \{\mathcal{D}_l, \mathcal{D}_u\}$ be a dataset associated with these tasks; it consists of $n$ labeled data samples $\mathcal{D}_l = \{(x_i, y_i)\}_{i=1}^n$ and $m$ unlabeled samples $\mathcal{D}_u = \{x_i\}_{i=1}^m$, where $m \gg n$, feature $x_i \in \mathcal{X}$, and label $y_i \in \mathcal{Y} = \{1, ..., N\}$. Under this semi-supervised continual learning, $\mathcal{D}_l$ is divided into multiple task sets $\mathcal{D}_l = \cup_{i \in \{1...k\}} \mathcal{D}_l^i$ based on labels (e.g., dividing CIFAR-10 dataset

into 5 tasks with two labels associated with each task). For each task $T_i$, we can only access labeled samples from a subset $\mathcal{D}_l^i \subset \mathcal{D}_l$ and unlabeled samples from $\mathcal{D}_u$.

We shall consider semi-supervised continual learning in an open environment, where unlabeled data $x \in \mathcal{D}_u$ may come from the known (seen) classes $C_l$ in labeled dataset $\mathcal{D}_l$ or novel, never-before-seen classes (OOD data) $C_n$, i.e., unlabeled data $\mathcal{D}_u$ is from classes $C_u = C_l \cup C_n$. In the context of continual learning, known classes $C_l$ in $\mathcal{D}_l$ are divided into $\{C_l^1, ..., C_l^k\}$, with $C_l^i \cap C_l^{i+1} = \emptyset$. Because the number of known classes is increasing along with task change in continual learning. We further denote known classes $C_s^i = \cup_{j=1}^i C_l^j$ up to task $T_i$ as the *task seen classes*, and the $C_n^i = C_u \backslash C_s^i$ as *task unseen classes*.

The goal is to continuously learn a model $f$ from a sequence of tasks $T$ that (i) can learn from novel classes and identify them, and (ii) correctly classify known classes while avoiding forgetting the previously learned tasks as the model gets updated. To achieve this, we seek to minimize the open risk (Scheirer et al., 2014) under continual learning constraints (Lopez-Paz & Ranzato, 2017):

$$f_t = \arg\min_{f \in \mathcal{H}} \ R\left(f(\mathcal{D}_l^t)\right) + \bar{\lambda} R_{\mathcal{O}_t}(f) \tag{1}$$

$$\text{s.t.} \ R\left(f_t(\mathcal{D}_l^i)\right) \leq R\left(f_{t-1}(\mathcal{D}_l^i)\right); \forall i \in [0...t-1]$$

where $R\left(f(\mathcal{D}_l^t)\right)$ denotes the empirical risk of $f$ on *known* training data at task $t$. $f_t$ is the model learned at the end of task $t$; $R_{\mathcal{O}_t}(f)$ is the *open space risk* (Scheirer et al., 2012) and is defined as

$$R_{\mathcal{O}_t}(f) = \frac{\int_{\mathcal{O}_t} f(x)\mathrm{d}x}{\int_{\mathcal{S}} f(x)\mathrm{d}x}.$$

where $\mathcal{S}$ is a space containing all ID samples and OOD samples that are mislabeled as ID. These OOD samples formulate an open space $\mathcal{O}$ in the $\mathcal{S}$. $R_{\mathcal{O}_t}(f)$ measures the potential risk of a function $f$ misclassifying samples that are in open space $\mathcal{O}_t$. Hyper-parameter $\lambda \geq 0$ is a regularization constant. Under the constraint in equation 1, the model performance on known classes does not decrease as the model gets updated.

## 4 PROPOSED METHOD

In this section, we introduce *OpenACL* which includes three main components: (i) prototype distance learning; (ii) semi-supervised prototype representation learning; and (iii) prototype adaptation. Specifically, *prototype distance learning* ensures that labeled data are associated with corresponding prototypes while maximizing their similarity. *Semi-supervised prototype representation learning* enhances model robustness against distribution shifts across tasks over time; it encourages samples with intrinsic similarities to converge to shared prototypes. *Prototype adaptation* identifies and allocates the most suitable prototypes from the prototype pool for an incoming task to facilitate the model's adaptation to the new task.

### 4.1 CONTINUAL PROTOTYPE LEARNING OVERALL

The key insight in Open SSCL is exploiting data from open-world datasets to improve the adaptation ability on new tasks of continual models. Instead of discarding OOD samples, we treat them as potential future ID data in upcoming tasks. In particular, we propose a novel continual prototype learning mechanism that mines the intra-class patterns in both labeled and unlabeled data. Consequently, data samples with similar representations are under a shared prototype. We shall maintain "seen prototypes" for known classes and "novel prototypes" for potential future classes. These novel prototypes can capture the patterns of classes in future tasks even before they are officially labeled. This proactive approach gives OpenACL an advantage, readying it to quickly adapt to new tasks.

### 4.2 PROTOTYPE DISTANCE LEARNING

Let $\mathcal{G} = \{g_1 \ldots g_m\}$ be denoted as $m$ prototypes for seen and novel classes, and let $h$ be the function that maps data to the representation. Unlike most existing Prototype Learning (PL) methods that compute the prototypes as the geometric centers of representations, our method models these prototypes as *trainable parameters*. This is because geometric centers are typically computed by

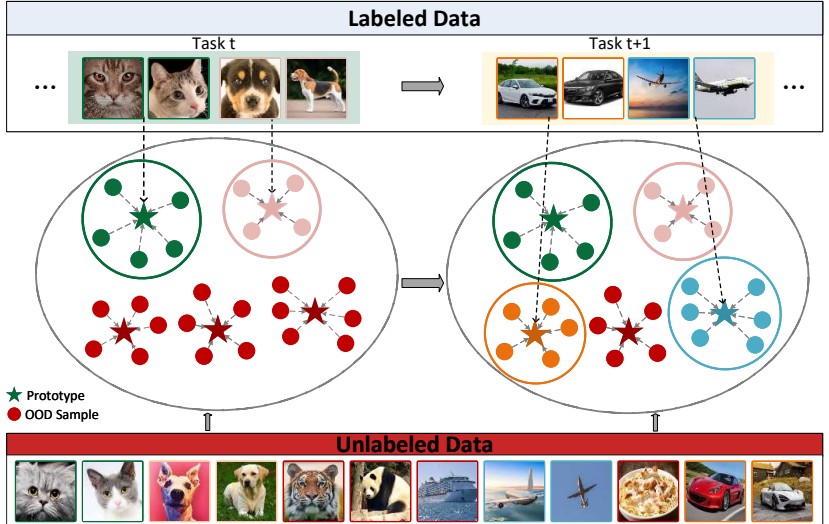

Figure 1: In OpenACL, we minimize the distance between data representations of seen classes and their labeled prototypes. Concurrently, semi-supervised prototype contrastive learning encourages similar representations to share the same prototype distribution and enhances representations for both known and novel classes. We assume data from the same class have similar representations in the latent space, so the novel prototypes are used to cluster representations from novel classes. Upon entering the adaptation phase for a new task $t + 1$, we receive labeled data in task $t + 1$. For each class within the task, we identify the prototype from the prototypes pool that is the most analogous and allocate the class label to it. By assigning novel prototypes to incoming task classes, we could have some well-trained prototypes and speeding up the learning process for new task prototypes.

averaging the data points from the same class; for unknown classes, the absence of labels makes it unclear which data points should be grouped together to average. In contrast, a trainable prototype can be learned to capture the pattern for a novel class. In our approach, the cosine distance is used to measure the distance between prototypes $g$ and representation $h(x)$ and we want to minimize the distance between the data representation and its corresponding class prototype.

Formally, for the labeled dataset $\mathcal{D}_l = \{(x_i, y_i)\}_{i=1}^n$ when the ground truth of data $x_i$ is known, our objective is to maximize the cosine similarity $sim(g_{y_i}, h(x_i)) = \frac{g_{y_i}^T h(x_i)}{||g_{y_i}|| \cdot ||h(x_i)||}$. We thus define the loss function $\mathcal{L}_p$ that encourages the data to be closer to its class prototype at task $t$ as:

$$\mathcal{L}_p = -\frac{1}{|B_l|} \sum_{i=1}^{|B_l|} \log \frac{\exp\left(sim\left(g_{y_i}, h(x_i)\right) \times s\right)}{\sum_{j=1}^{|\mathcal{G}|} \exp\left(sim\left(g_j, h(x_i)\right) \times s\right)} \qquad (2)$$

In equation 2, $|B_l|$ is the number of labeled samples in a batch $B_l$. The hyper-parameter $s$ controls the softmax temperature when transforming similarity into probability, ensuring stable training (Wang et al., 2018). By minimizing $\mathcal{L}_p$, we align a representation with its class prototype while distancing it from other prototypes.

### 4.3 SEMI-SUPERVISED PROTOTYPE REPRESENTATION LEARNING

To equip the prototypes with the ability to exploit the open-world data and represent novel classes, we introduce semi-supervised prototype contrastive learning to learn powerful representations for both unlabeled and labeled data and assign data with similar representations to a common prototype. Contrastive learning is designed to extract meaningful representations by exploiting both the similarities and dissimilarities between data instances. This is typically achieved by comparing two augmented views (e.g., rotation, flipping, resizing) of the same instance or different instances. However, rather than solely maximizing the alignment between different views of an instance in latent space, our objective is to maintain consistency in the distribution of these representations over the

prototypes. Put differently, our aim shifts to amplifying the consistency of prototype distributions rather than focusing exclusively on representation.

Given an instance $x$, we first generate two augmented views $\tilde{x}$ and $\tilde{x}'$ and obtain their representations $h(\tilde{x})$ and $h(\tilde{x})'$ as suggested in (Chen et al., 2020). The probability of a view $\tilde{x}$ being assigned to a prototype $g_i$ can be computed as:

$$p_i(\tilde{x}) = \frac{\exp\left(sim\left(g_i, h(\tilde{x})\right) \times s\right)}{\sum_{j=1}^{|\mathcal{G}|} \exp\left(sim\left(g_j, h(\tilde{x})\right) \times s\right)} \tag{3}$$

Then we can align the distribution to prototypes between two views via a contrastive loss:

$$\mathcal{L}_c^u = -\frac{1}{|\mathcal{B}_u|} \sum_{i=1}^{|\mathcal{B}_u|} \log \frac{\exp(sim(p(\tilde{x}_i), p(\tilde{x}_i'))/\kappa)}{\sum_{j=1}^{|\mathcal{B}_u|} \mathbf{1}_{[x_j \neq x_i]} \exp(sim(p(\tilde{x}_i), p(\tilde{x}_j))/\kappa)} \tag{4}$$

where $\mathcal{B}_u$ is an unlabeled minibatch including pairs of two augmented views $\tilde{x}$ and $\tilde{x}'$ from $x$. $\kappa$ is a temperature parameter. $\mathbf{1}_{[\cdot]} \in \{0, 1\}$ is the condition function. To further enhance the power and robustness of our representations, we also leverage labeled data to extend the unsupervised prototype contrastive learning to semi-supervised prototype contrastive learning. This is advantageous as the labeled data can provide direct information about the relationship between instances and their corresponding prototypes. Following Khosla et al. (2020), we incorporate supervised contrastive learning in our prototype representation learning. For labeled minibatch $\mathcal{B}_l$ and unlabeled minibatch $\mathcal{B}_u$ with two augmented views, we have a conjunct contrastive loss on prototype distribution:

$$\mathcal{L}_c = \mathcal{L}_c^u - \sum_{i=1}^{|\mathcal{B}_l|} \log \frac{1}{|P_i|} \sum_{\tilde{x}_j \in P_i} \frac{\exp(sim(p(\tilde{x}_i), p(\tilde{x}_j))/\kappa)}{\sum_{\tilde{x}_k \in A(i)} \exp\left(sim(p(\tilde{x}_i), p(\tilde{x}_k))/\kappa\right)} \tag{5}$$

Here, $A(i)$ is a set $\mathcal{B}_l \setminus \{\tilde{x}_i\}$. $P_i$ is the set of all positive samples $\{\tilde{x}_j \in A(i) : y_j = y_i\}$.

The final objective function combines both the contrastive loss and the supervised loss, weighted by a hyper-parameter $\lambda$, i.e., the loss at task $t$ is $\mathcal{L} = \mathcal{L}_p + \lambda \mathcal{L}_c$. We set $\lambda$ as 1 in our method.

The rationale of the prototype-level contrastive learning mechanism is straightforward: if two representations are close in the latent space, they will also have similar distributions over the prototypes. Based on the assumption that data from the same class should have similar representation in latent space, instances being pushed closer to the same prototype can be classified in a class. By specifying the prototypes for novel classes, we reduce the intra-class variance (pushing similar instances towards these prototypes) to decrease the model's tendency to misclassify OOD samples in the seen classes, thereby decreasing the open space risk $R_{\mathcal{O}_t}$. In equation 5, unlabeled data also includes data in previous trained tasks. Thus, our model could leverage a comprehensive prototype representation that spans previous tasks, the current task, and future tasks to ensures a consistent representation space for continual learning. This inherently provides a regularizing effect to make up for catastrophic forgetting, minimizing the risk of overwriting previous information.

### 4.4 PROTOTYPE ADAPTATION

The aforementioned prototype learning establishes a set of novel prototypes learned from the intra-class similarities within the unlabeled data. These novel prototypes can be further used in CL to adapt to the new task. Intuitively, upon transitioning from task $t$ to the subsequent task $t + 1$, the classes in the forthcoming task should already possess associated prototypes, courtesy of their presence in the unlabeled data. Thus, we could associate these potential prototypes with new classes and adapt the model to the task $t + 1$ quickly.

Specifically, consider labeled data $\{(x, y) \in \mathcal{D}_l^{t+1}\}$ at new task $t+1$. For each class label $\bar{y} \in C_l^{t+1}$, we want to find the most potential prototype for class $\bar{y}$. Define a count function $I(x, g_j)$ that returns 1 if $g_j$ is the most similar prototype for $x$ and 0 otherwise:

$$I(x, g_j) = \begin{cases} 1 & \text{if } g_j = \arg\max_{g_k \in \mathcal{G}} sim(x, g_k) \\ 0 & \text{otherwise} \end{cases} \tag{6}$$

We then determine the prototype $g_{\bar{y}}^* \in \mathcal{G}$ by the number of its closet samples in $\{(x, y) \in \mathcal{D}_l^{t+1} : y = \bar{y}\}$. The one with the most grouped samples will be selected as $g_{\bar{y}}^*$ and be assigned with label $\bar{y}$.

$$g_{\bar{y}}^* = \arg\max_{g_j \in \mathcal{G}} \sum_{(x_i, y_i) \in \mathcal{D}_l^{t+1} : y_i = \bar{y}} I(x_i, g_j) \tag{7}$$

In implementation, if multiple classes are associated with the same prototype, we randomly assign a class label $y$ from these classes to the prototype. In addition, to avoid the trivial solution that all instances are assigned to a single prototype in the early stage of the training (Caron et al., 2018; Cao et al., 2022), we adopt a reinitialization strategy. After assigning labels for task $t + 1$ but before entering its training, the unassigned novel prototypes are reinitialized. To establish these initial novel prototypes as a new task begins, we deploy the $K$-means algorithm, using cosine distance as a metric to cluster centroids as initial novel prototypes. The known prototypes are used as prior knowledge for the $K$-means algorithm, but remain static and are not subjected to updates post-clustering. Specifically, given the prototype pool $\mathcal{G}$ and the set of seen class prototypes for $C_s^{t+1}$, the initialized centroids in $K$-means algorithm are selected as $|C_s^{t+1}|$ known prototypes and $|\mathcal{G}| - |C_s^{t+1}|$ randomly selected data points from the unlabeled dataset $D_u$. To reduce computation cost, $K$-means is running on a subset of $D_u$ to obtain $|\mathcal{G}|$ centroids. we identify $|C_s^{t+1}|$ centroids that are most similar to the known prototypes and exclude them using cosine similarity. The remaining centroids are used to initialize the novel prototypes in the prototype pool. This ensures a more representative set of prototypes for subsequent tasks and solves the trivial solution problem.

## 5 EXPERIMENTS

In this section, we introduce the datasets and the baselines. Then, we present results from various benchmarks in comparison to baselines. Implementation details are available in Appendix A.1.1.

### 5.1 EXPERIMENT SETTING

**Datasets.** We adopt the following datasets in experiments. The data from known classes is partitioned into labeled and unlabeled segments with ratios of 20% labeled data and 50% labeled data.

1. **CIFAR-10 (Krizhevsky et al., 2009):** The first 6 classes are organized into 3 tasks ($k = 3$), each containing two classes. The remaining 4 classes are treated as unknown. For each task, we have 2,000 labeled instances under the 20% split and 5,000 labeled instances under the 50% split.
2. **CIFAR-100 (Krizhevsky et al., 2009):** The initial 80 classes from CIFAR-100 are segmented into 16 tasks ($k = 16$). The subsequent 20 classes are treated as unknown. For every task, 500 instances are labeled under the 20% split, and 1,250 instances are labeled under the 50% split.
3. **Tiny-ImageNet (Deng et al., 2009):** The initial 120 classes of Tiny-ImageNet are categorized into 20 tasks ($k = 20$), leaving 80 classes as unknown. For each task, there are 600 labeled instances in the 20% split and 1,500 labeled instances in the 50% split.

Using the above split, we take two datasets as input: labeled $D_l = \{\mathcal{D}_l^1, ..., \mathcal{D}_l^k\}$ and unlabeled $D_u$ consisting of unlabeled data from **known classes** $C_l$ and all data from **unknown classes** $C_n$. For each task $i$, we simultaneously sample data from the $\mathcal{D}_l^i$ for the current task and the $D_u$. The proportion of labeled to unlabeled data in the sample matches the respective proportions in the datasets. Note that, as $D_u$ is randomly shuffled, we can access all classes in $D_u$ in each task. We sequentially sampled the data from the $D_u$ without knowing the source, i.e., the data comes from previous task classes, current task classes, future task classes, and OOD classes (unknown classes).

**Baselines.** We compare our algorithm (OpenACL) with existing methods in CL in both *task incremental learning (Task-IL)* and *class incremental learning (Class-IL)* settings. The distinction between these settings is elaborated upon in Appendix A.1.1. Additionally, our focus is on online continual learning, where models are only allowed to be trained for 1 epoch. To ensure a fair comparison, we first equip supervised learning-based methods with a well-known semi-supervised learning method: FixMatch (Sohn et al., 2020). Then, as our prototypes use the contrastive learning idea to align the distribution, we also add contrastive learning loss (Chen et al., 2020) to baselines to learn representation from unlabeled data. These baselines include:

1. *Single* (Lopez-Paz & Ranzato, 2017): It sequentially trains a single network across all tasks.
2. *Independent* (Lopez-Paz & Ranzato, 2017): It trains multiple networks; each is trained independently on data from one task.
3. *EWC* (Kirkpatrick et al., 2017): Elastic Weight Consolidation (EWC) is a regularization technique that adds a penalty loss function to minimize the changes in the weights that are important for previous tasks while still allowing the weights to be updated for new tasks.

4. *GEM* (Lopez-Paz & Ranzato, 2017): Gradient Episodic Memory (GEM) maintains an episodic memory to store samples from previous tasks and ensure the gradients for new tasks do not interfere with learned tasks.

5. *iCaRL* (Rebuffi et al., 2017): iCaRL uses a nearest-exemplar method and distillation to maintain a set of exemplars for each class.

6. *GSS* (Aljundi et al., 2019): Gradient-based sample selection(GSS) selects and replays a subset of diverse data based on the gradient to solve online continual learning.

7. *ER* (Chaudhry et al., 2019): Experience Replay (ER) trains both incoming data and data from the replay memory. Despite its simplicity, ER surpasses many advanced continual learning methods.

8. *DER* (Buzzega et al., 2020): Dark Experience Replay(DER) stores examples with their outputs, and minimizes the difference between outputs from the current model and memory.

9. *DistillMatch* (Smith et al., 2021): DistillMatch is distillation-based that considers SSCL by rejecting OOD samples. It uses each data more than once to train the model and OOD detector. To adapt DistillMatch to online continual learning, we provide the ground truth for OOD samples, assisting in their exclusion.

Table 1: Average accuracy over three runs of experiments on Task-IL benchmarks. Some baselines are adapted to SSL by incorporating them with FixMatch (Sohn et al., 2020) or SimCLR (Chen et al., 2020) to learn from unlabeled data. Results are organized as SimCLR usage / FixMatch usage / No unlabeled data usage. The standard deviation results are reported in the Appendix A.3.

| Method | CIFAR-10 | | CIFAR-100 | | Tiny-ImageNet | |
|---|---|---|---|---|---|---|
| Labels % | 20 | 50 | 20 | 50 | 20 | 50 |
| Single | 57.5 / 57.6 / 54.7 | 59.3 / 57.0 / 57.6 | 33.5 / 34.1 / 32.3 | 37.9 / 36.3 / 37.2 | 20.9 / 20.5 / 19.6 | 25.9 / 23.3 / 23.1 |
| Independent | 62.5 / 64.2 / 61.3 | 63.9 / 62.3 / 62.5 | 26.7 / 30.3 / 31.8 | 36.2 / 36.2 / 33.4 | 21.6 / 21.5 / 23.2 | 26.5 / 28.0 / 27.0 |
| EWC | 57.1 / 56.1 / 58.9 | 57.8 / 56.3 / 59.2 | 33.4 / 34.9 / 33.8 | 35.8 / 35.6 / 36.3 | 20.0 / 20.2 / 19.8 | 25.0 / 22.4 / 24.1 |
| iCaRL | 56.0 / 57.4 / 56.7 | 57.2 / 58.7 / 58.3 | 45.8 / 45.9 / 46.4 | 44.1 / 42.3 / 41.8 | 25.2 / 25.3 / 23.5 | 31.3 / 29.0 / 26.5 |
| DER | 62.2 / 63.9 / 63.3 | 63.2 / 63.9 / 63.6 | 38.6 / 38.7 / 39.6 | 46.8 / 44.7 / 44.0 | 24.2 / 22.4 / 25.8 | 28.4 / 29.6 / 28.0 |
| GEM | 61.3 / 64.0 / 62.6 | 63.2 / 63.6 / 64.2 | 53.5 / 52.6 / 51.8 | 58.6 / 57.5 / 54.4 | 33.0 / 35.4 / 32.1 | 40.1 / 37.3 / 38.0 |
| ER | 62.9 / 62.3 / 61.3 | 64.9 / 63.8 / 62.6 | 54.8 / 55.3 / 53.7 | 59.9 / 58.5 / 57.8 | 35.2 / 36.3 / 35.7 | 41.7 / 41.4 / 40.2 |
| DistillMatch | 57.8 | 59.4 | 35.7 | 41.3 | 21.8 | 26.2 |
| **OpenACL** | **64.3** | **66.3** | **60.4** | **66.6** | **40.2** | **47.0** |

Table 2: Average accuracy over three runs of experiments on Class-IL benchmarks.

| Method | CIFAR-100 | | Tiny-ImageNet | |
|---|---|---|---|---|
| Labels % | 20 | 50 | 20 | 50 |
| Single | 3.1 / 2.8 / 2.5 | 3.0 / 2.5 / 3.0 | 1.9 / 2.0 / 1.7 | 2.4 / 2.8 / 2.7 |
| iCaRL | 6.8 / 7.0 / 6.3 | 7.3 / 8.3 / 7.0 | 4.5 / 3.3 / 3.4 | 4.1 / 4.8 / 4.2 |
| DER | 3.7 / 3.7 / 3.5 | 3.6 / 3.9 / 3.9 | 2.4 / 2.5 / 2.1 | 2.4 / 2.6 / 2.3 |
| GEM | 7.0 / 8.0 / 6.9 | 9.7 / 7.7 / 6.7 | 2.4 / 3.4 / 2.7 | 2.3 / 2.6 / 1.8 |
| GSS | 12.8 / 11.2 / 10.3 | 16.8 / 15.3 / 15.2 | 3.3 / 5.4 / 3.8 | 5.3 / 5.6 / 5.0 |
| ER | 10.9 / 12.0 / 11.5 | 15.6 / 15.8 / 16.9 | 3.3 / 4.2 / 3.9 | 4.8 / 6.7 / 5.7 |
| DistillMatch | 2.8 | 3.2 | 2.0 | 2.7 |
| **OpenACL** | **15.7** | **20.0** | **7.9** | **11.9** |

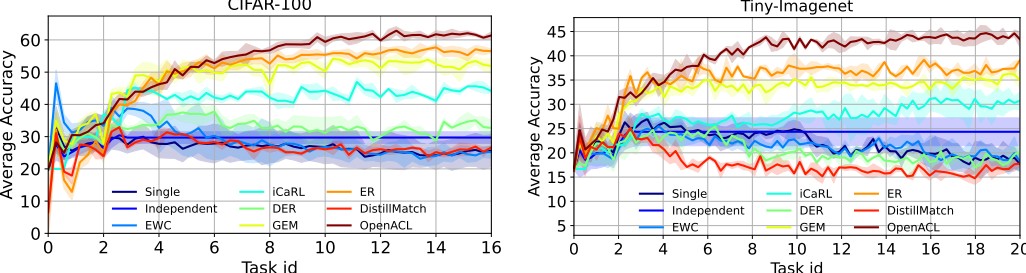

Figure 2: Average accuracy of the first three tasks on 50% labeled CIFAR-100 and Tiny-ImageNet during Task-IL training. We test the models on the first three tasks after finishing subsequent tasks to examine their ability to preserve prior knowledge.

## 5.2 RESULTS

**Evaluation on split datasets.** We contrasted our algorithm against established baselines in the online Task-IL setting and online Class-IL setting with varying label ratios across seen classes. To make a fair comparison, supervised continual learning methods are integrated with FixMatch or SimCLR. Table 1 and 2 present the mean accuracy across all tasks for each method, both with and

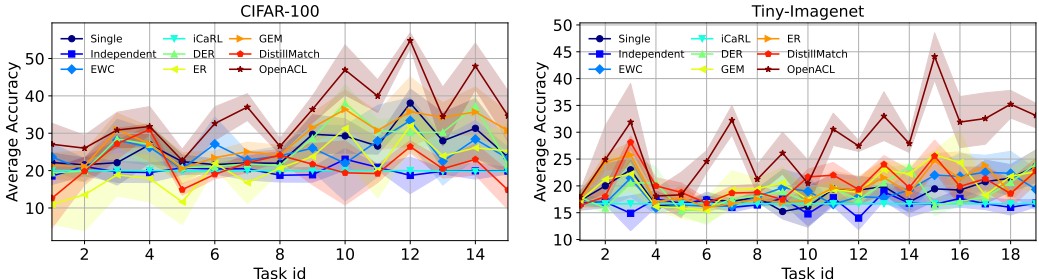

Figure 3: Average accuracy on a novel task after training with a single batch in Task-IL.

Table 3: BWT and FWT results on 50% labeled dataset. We report the best results among three implementations(SimCLR, FixMatch, and Normal). The results show as BWT / FWT.

|  | Single | Independent | EWC | iCaRL | DER | GEM | ER | DistillMatch | OpenACL |
|---|---|---|---|---|---|---|---|---|---|
| CIFAR-100 | -5.3 / 0.9 | 0 / 0 | -5.7 / 0.4 | -5.3 / 0 | 0.3 / -0.3 | **11.6** / -0.3 | 11.5 / -5.1 | -6.5 / -1.8 | 9.2 / **13.0** |
| Tiny-ImageNet | -6.3 / 0.4 | 0 / 0 | -8.6 / 0.6 | -1.1 / 0 | -0.5 / 0.8 | **4.8** / 0.1 | 4.3 / 0.6 | -11.8 / -0.1 | 2.7 / **10.9** |

without the inclusion of unlabeled data. The results in the Task-IL setting and Class-IL setting demonstrate that OpenACL outperforms all baselines on all datasets, with significant margins in most cases. Notably, we observe that some baselines also benefit from unlabeled data enhanced by FixMatch or SimCLR. This emphasizes the potential benefits of unlabeled data in the context of CL. However, directly integrating CL with unlabeled data usage yields only modest improvements, highlighting the need for more specialized methods for Open SSCL, like OpenACL. OpenACL's superior performance suggests that specialized algorithms tailored for Open SSCL can provide significant benefits over traditional methods or straightforward combinations of the existing methods.

**Mitigate catastrophic forgetting.** We follow Lopez-Paz & Ranzato (2017) to compare backward transfer (BWT) and forward transfer (FWT) in Table 3. Positive BTW suggests that performance on old tasks improved after learning new tasks, while a negative BWT implies that the model forgot some of the previous tasks. GEM, which requires gradient constraint achieves the best BWT among these baselines, while OpenACL achieves comparable performance as GEM and ER on solving catastrophic forgetting. We also track the average test accuracy on the first three tasks over time to examine catastrophic forgetting. The results are presented in Figure 2. It shows that our method performs the best on the first three tasks during training and is also more stable than baselines. Besides, along with training, OpenACL even achieves better performance on the first few tasks, while some baselines almost forget the first three tasks completely, especially in challenging datasets like Tiny-ImageNet. These results validate that OpenACL can help to tackle catastrophic forgetting.

**Adaptability to new tasks.** FWT in Table 3 indicates the effect on the performance of learning new tasks from prior learning. A positive FWT suggests the model's "zero-shot" learning ability for unseen tasks. The results show that OpenACL exhibits superior performance in FWT, highlighting its exceptional zero-shot learning capability, confirming that it can swiftly adapt to new tasks leveraging unlabeled data knowledge. Further underlining its adaptability, we investigate the adaptability by comparing accuracy after training a single batch of data in a new task. Figure 3 shows OpenACL attains high accuracy across all tasks and maintains a stable performance throughout the process, suggesting that our algorithm can efficiently learn and adapt to new tasks.

# 6 CONCLUSION

In this paper, we study continual learning in an open scenario and formulate open semi-supervised continual learning (Open SSCL). Distinct from traditional, Open SSCL learns from both labeled and unlabeled data and allows novel classes to appear in the unlabeled dataset. Recognizing the relationship between transitions from known tasks to upcoming tasks in CL and shifts from ID classes to OOD classes, we propose a prototype-based approach: OpenACL. OpenACL exploits the open-world data to enhance the adaptability of continual learning models, while simultaneously mitigating catastrophic forgetting. Our study highlights the importance of using unlabeled data and novel classes in CL and the potential of Open SSCL as a promising direction for future research.

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

# A  APPENDIX

## A.1  ADDITION EXPERIMENT SETTING

### A.1.1  IMPLEMENTATION DETAILS

All experiments are conducted on a server equipped with multiple NVIDIA V100 GPUs, Intel Xeon(R) Platinum 8260 CPU, and 256GB memory. The code is implemented with Python 3.9 and PyTorch 1.10.0.

We used the same network architecture as (Lopez-Paz & Ranzato, 2017), a reduced ResNet18 for CIFAR and Tiny-ImageNet images. We consider two settings: *task incremental learning (Task-IL)* and *class incremental learning (Class-IL )*. Task-IL assumes task id is known and used to select a classifier (separate logits) for a specific task, while it is not allowed to use task id in Class-IL. Therefore the Class-IL setting is much more challenging than the Task-IL setting. Note that, OpenACL only uses the task id to separate logits for $\mathcal{L}_p$ in the Task-IL setting. In addition, the online training setting is used in our experiments where the model is only allowed to train 1 epoch on task data, every labeled and unlabeled sample is only seen once. However, we also perform 3 iterations over a batch in Class-IL following Aljundi et al. (2019). Note that, it is different from training multiple epochs on a task.

We train models using a stochastic gradient descent (SGD) optimizer. For replay-based methods, the size of the replay memory is set to 250 per task under 50% labeled dataset and 125 per task under 20% labeled dataset. In the Task-IL setting, we allow the use of task id to separate the replay memory. Therefore, we maintain task-specific replay memories for baselines like DER and ER. OpenACL is also equipped with a simple replay memory using Reservoir Sampling (Vitter, 1985) to store labeled data. At every iteration, we retrieve samples from the replay memory to update the model using equation 2. The number of drawn samples from replay memory is fixed to 10 in Task-IL and 30 in Class-IL. However, GEM still uses the full memory. The hyperparameters for baselines are set to the suggested value in their original implementation after grid search. In all experiments, we set the batch size for labeled data to 10, and the batch size of unlabeled data to $10 \cdot \frac{\mathcal{D}_u}{\mathcal{D}_l}$. It ensures that the ratio of unlabeled to labeled data in each batch is proportionate to their overall distribution in the datasets.. The temperature $s$ in equation 2 and equation 3 is set to 10, as suggested in previous methods (Cao et al., 2022), and $\kappa$ is set to 0.07 as the original setting in (Chen et al., 2020). We use 50% of the unlabeled dataset to run the $K$-mean during the prototype adaptation. Grid search is used to find the best learning rate for baselines, searching from [0.001, 0.01, 0.05, 0.1, 0.5, 1.0]. The threshold in FixMatch of baselines is set to 0.8.

### A.1.2  METRIC

Three metrics are used in our experiments, including Accuracy (ACC), Backward Transfer (BWT), and Forward Transfer (FWT) (Lopez-Paz & Ranzato, 2017; Yan et al., 2021).

**ACC:** We report the average accuracy on all trained tasks to evaluate the fundamental classification performance of all methods.

**BWT:** BWT measures the influence of learning a new task $t$ on previous tasks $\{1, ..., t-1\}$. To calculate the BWT, we define accuracy on test classes $C_l^t$ at task $t$ as the $A_{C_l^t}^t$. BWT is computed as follows:

$$\text{BWT} = \frac{1}{|T-1|} \sum_{i=2}^{|T|} \frac{1}{i} \sum_{j=1}^{i} A_{C_l^i}^i - A_{C_l^j}^j \tag{8}$$

**FWT:** FWT gauges how the model performs on upcoming task $t+1$ at task $t$. Let $\bar{a}$ be a vector storing accuracy for all tasks at random initialization status. After finishing all the tasks, we have FWT:

$$\text{FWT} = \frac{1}{|T-1|} \sum_{i=2}^{|T|} A_{C_l^i}^{i-1} - \bar{a}_i \tag{9}$$

### A.1.3 DATASET ILLUSTRATION

In this section, we provide a more detailed illustration of our datasets.

The CIFAR-10 dataset comprises 50,000 images across 10 classes. We designate the first 6 classes as seen classes and divide them into 3 tasks, each encompassing 2 classes. For these 6 classes, we further split data from them into labeled and unlabeled subsets. In our experiment, we adopt two different division ratios for data from seen classes: 20% labeled (thus, 80% unlabeled) and 50% labeled (equally, 50% unlabeled). For example, with a 20% labeling ratio, each class includes 1,000 labeled and 4,000 unlabeled instances, so $|D_l|$ is 6,000. We then maintain the unlabeled dataset $D_u$ using the unlabeled instances from the seen classes and all data from the 4 unknown classes, totaling 44,000 instances. Similarly, with a 50% labeling ratio, each class has 2,500 labeled and 2,500 unlabeled instances, leading to $D_l$ with 15,000 labeled instances and $D_u$ with 35,000 unlabeled instances.

For the CIFAR-100 dataset, which includes 50,000 images across 100 classes, the first 80 classes are treated as seen classes and divided into 16 tasks with five classes each. Under a 20% labeled and 80% unlabeled ratio, there are 8,000 labeled instances and 32,000 unlabeled instances across 80 seen classes. The corresponding unlabeled dataset $D_u$ consists of 32,000 unlabeled instances from 80 seen classes and 10,000 instances from 20 unknown classes.

The Tiny ImageNet contains 100000 images of 200 classes (500 for each class). We split the first 120 classes into 20 tasks, each containing 6 classes. Under a 20% labeled and 80% unlabeled ratio, we have 12,000 labeled instances and 48,000 unlabeled instances. The unlabeled dataset $D_u$ consists of 48,000 unlabeled instances from 120 seen classes and 40,000 instances from 80 unknown classes.

During training, for each task $i$, we simultaneously sample data from the labeled dataset $\mathcal{D}_l^i$ for the current task $i$ and the shuffled unlabeled dataset $D_u$. $D_u$ consists of data from all classes, including previous task classes, current task classes, future task classes (whose labels have not been revealed and are thus treated as OOD for the current task $i$), and unknown (OOD) classes that are not included in the continual learning tasks. In each iteration, we sample both labeled and unlabeled data for each batch, adhering to the respective proportions of labeled and unlabeled data in the datasets. For example, in the CIFAR-10 dataset with a 50% labeling ratio, where we have 15,000 labeled instances and 35,000 unlabeled instances, we maintain this proportion in our sampling approach for each iteration. Consequently, in a single batch, we sample 10 labeled instances and 23 unlabeled instances. For each task, we access 5,000 labeled instances from 2 classes, and 11,500 instances from 10 classes. This approach ensures that each unlabeled data is utilized only once in the online continual learning process.

### A.2 ABLATION STUDY

### A.2.1 ABLATION ON ADAPTATION

In this section, we conduct an ablation study on the CIFAR-100 and Tiny-ImageNet datasets by removing each component separately to examine their importance. Specifically, we systematically evaluate the impact of (i) Omitting the prototype adaptation (denoted as w/o PA), (ii) Excluding the $k$-means initialization in the prototype adaptation (denoted as w/o K), (iii) Omitting prototype allocation for new tasks while retaining the $k$-means initialization in the prototype adaptation (denoted as w/o A). The analysis of w/o PA is intended to explain the effectiveness of prototype adaptation when shifting to new tasks. Meanwhile, the evaluation of w/o K aims to affirm that the model's adaptability is mainly from our continual prototype learning mechanism, not the $k$-means initialization. OpenACL w/o A is discussed to show the sole influence of the $k$-means initialization.

As shown in Table 4, the performance of OpenACL is compromised upon the removal of any single component. We mainly consider FWT in this experiment because the prototype adaptation is designed to adapt to the new tasks. A comparison between OpenACL w/o PA and OpenACL demonstrates a considerable enhancement in FWT with the use of the prototype adaptation. However, even without the prototype adaptation, the model still manages a mild positive FWT which verifies that our method can learn a general representation for both ID samples and OOD samples.

Furthermore, it also shows that the improvement of adaptation is not achieved by $k$-means initialization. By looking at OpenACL w/o K, it still achieves good performance on FWT compared

Table 4: Ablation study on the prototype adaptation. We report average accuracy over three runs using different variants of OpenACL in Task-IL.

| | CIFAR-100 | | | Tiny-ImageNet | | |
|---|---|---|---|---|---|---|
| | ACC | BWT | FWT | Acc | BWT | FWT |
| w/o PA | $65.9_{\pm 0.77}$ | $10.1_{\pm 1.65}$ | $0.4_{\pm 3.40}$ | $45.6_{\pm 0.22}$ | $2.8_{\pm 0.15}$ | $0.7_{\pm 0.70}$ |
| w/o K | $66.2_{\pm 0.94}$ | $10.2_{\pm 1.49}$ | $9.8_{\pm 1.13}$ | $46.2_{\pm 0.36}$ | $3.4_{\pm 1.54}$ | $9.9_{\pm 0.38}$ |
| w/o A | $66.4_{\pm 0.38}$ | $7.6_{\pm 0.99}$ | $1.4_{\pm 1.01}$ | $45.1_{\pm 0.38}$ | $1.9_{\pm 1.01}$ | $1.0_{\pm 0.90}$ |
| OpenACL | $66.6_{\pm 0.28}$ | $9.2_{\pm 1.65}$ | $13.0_{\pm 1.48}$ | $47.0_{\pm 0.42}$ | $2.7_{\pm 1.36}$ | $10.9_{\pm 1.10}$ |

with others. Therefore, $k$-means initialization is only used to amplify the adaptability of the model. Then, by analyzing the results of OpenACL w/o A, we could find that $k$-means initialization brings about a minor improvement but still serves a role in augmenting our adaptation strategy. In addition, ablation on the prototype adaptation also shows this component does not markedly affect accuracy.

### A.2.2 ABLATION ON UNLABELED DATA

To demonstrate the advantage of using unlabeled data, we compare OpenACL with its supervised learning counterpart, OpenACL(S). OpenACL(S) only uses eq equation 2 for model optimization without the use of unlabeled data, but keeps the prototype adaptation with the $k$-means initialization. The results are presented in Table 5. It is evident that, without using unlabeled data during training, the performance of OpenACL(S) aligns more closely with that of ER and GEM in terms of accuracy in table 1. Although OpenACL(S) retains some zero-shot learning capabilities, benefiting from the prototype adaptation, this ability is diminished with the exclusion of unlabeled data.

Table 5: Average accuracy of the ablation study, focusing on unlabeled data usage, across three runs on CIFAR-100 and Tiny-ImageNet within the Task-IL setting.

| | CIFAR-100 | | | Tiny-ImageNet | | |
|---|---|---|---|---|---|---|
| | ACC | BWT | FWT | Acc | BWT | FWT |
| OpenACL(S) | $58.8_{\pm 1.24}$ | $3.0_{\pm 0.57}$ | $7.8_{\pm 1.89}$ | $38.3_{\pm 1.12}$ | $-0.2_{\pm 0.49}$ | $4.1_{\pm 0.87}$ |
| OpenACL | $66.6_{\pm 0.28}$ | $9.2_{\pm 1.65}$ | $13.0_{\pm 1.48}$ | $47.0_{\pm 0.42}$ | $2.7_{\pm 1.36}$ | $10.9_{\pm 1.10}$ |

### A.3 SUPPLEMENTARY RESULTS

Here, we present the full version of table 1 and 2 in table 6 and 7.

Table 6: Table 1 with standard deviation

| Method | CIFAR-10 | | CIFAR-100 | | Tiny-ImageNet | |
|---|---|---|---|---|---|---|
| Labels % | 20 | 50 | 20 | 50 | 20 | 50 |
| Single | $57.5_{\pm 3.67}$ / $57.6_{\pm 3.49}$ / $54.7_{\pm 2.54}$ | $59.3_{\pm 2.78}$ / $57.0_{\pm 1.83}$ / $57.6_{\pm 2.05}$ | $33.5_{\pm 1.27}$ / $34.1_{\pm 3.10}$ / $32.3_{\pm 2.48}$ | $37.9_{\pm 2.82}$ / $36.3_{\pm 2.63}$ / $37.2_{\pm 1.61}$ | $20.9_{\pm 1.99}$ / $20.5_{\pm 0.69}$ / $19.6_{\pm 0.54}$ | $25.9_{\pm 1.14}$ / $23.3_{\pm 0.83}$ / $23.1_{\pm 1.16}$ |
| Independent | $62.5_{\pm 3.22}$ / $64.2_{\pm 1.35}$ / $61.3_{\pm 2.56}$ | $63.9_{\pm 3.49}$ / $62.3_{\pm 2.43}$ / $62.5_{\pm 2.83}$ | $26.7_{\pm 3.98}$ / $30.3_{\pm 3.28}$ / $31.8_{\pm 2.88}$ | $36.2_{\pm 2.30}$ / $36.2_{\pm 2.15}$ / $33.4_{\pm 1.67}$ | $21.6_{\pm 0.83}$ / $21.5_{\pm 1.07}$ / $23.2_{\pm 1.72}$ | $26.5_{\pm 0.84}$ / $28.0_{\pm 2.21}$ / $27.0_{\pm 1.79}$ |
| EWC | $57.1_{\pm 2.36}$ / $56.1_{\pm 3.10}$ / $58.9_{\pm 3.91}$ | $57.8_{\pm 2.57}$ / $56.3_{\pm 1.70}$ / $59.2_{\pm 2.79}$ | $33.4_{\pm 3.06}$ / $34.9_{\pm 4.06}$ / $33.8_{\pm 2.54}$ | $35.8_{\pm 1.20}$ / $35.6_{\pm 2.48}$ / $36.3_{\pm 1.31}$ | $20.0_{\pm 1.31}$ / $20.2_{\pm 1.48}$ / $19.8_{\pm 1.65}$ | $25.0_{\pm 1.29}$ / $22.4_{\pm 2.61}$ / $24.1_{\pm 1.42}$ |
| iCaRL | $56.0_{\pm 1.07}$ / $57.4_{\pm 1.38}$ / $56.7_{\pm 2.19}$ | $57.2_{\pm 1.35}$ / $58.7_{\pm 0.97}$ / $58.3_{\pm 2.20}$ | $45.8_{\pm 1.50}$ / $45.9_{\pm 2.68}$ / $46.4_{\pm 0.58}$ | $44.1_{\pm 1.38}$ / $42.3_{\pm 1.76}$ / $41.8_{\pm 1.09}$ | $25.2_{\pm 1.03}$ / $25.3_{\pm 1.75}$ / $23.5_{\pm 1.39}$ | $31.3_{\pm 1.01}$ / $29.0_{\pm 1.72}$ / $26.5_{\pm 2.71}$ |
| DER | $62.2_{\pm 0.71}$ / $63.9_{\pm 3.30}$ / $63.3_{\pm 2.00}$ | $63.2_{\pm 2.58}$ / $63.9_{\pm 2.42}$ / $63.6_{\pm 2.39}$ | $38.6_{\pm 3.03}$ / $38.7_{\pm 2.51}$ / $39.6_{\pm 3.24}$ | $46.8_{\pm 1.92}$ / $44.7_{\pm 2.36}$ / $44.0_{\pm 2.82}$ | $24.2_{\pm 2.64}$ / $22.4_{\pm 2.68}$ / $25.8_{\pm 1.02}$ | $28.4_{\pm 2.24}$ / $29.6_{\pm 2.27}$ / $28.0_{\pm 1.66}$ |
| GEM | $61.3_{\pm 1.08}$ / $64.0_{\pm 2.24}$ / $62.6_{\pm 2.18}$ | $63.2_{\pm 0.82}$ / $63.6_{\pm 2.39}$ / $64.2_{\pm 0.52}$ | $53.5_{\pm 1.38}$ / $52.6_{\pm 0.79}$ / $51.8_{\pm 0.82}$ | $58.6_{\pm 1.57}$ / $57.5_{\pm 1.59}$ / $54.4_{\pm 1.67}$ | $33.0_{\pm 1.07}$ / $35.4_{\pm 1.56}$ / $32.1_{\pm 1.49}$ | $40.1_{\pm 2.10}$ / $37.3_{\pm 1.20}$ / $38.0_{\pm 2.35}$ |
| ER | $62.9_{\pm 1.17}$ / $62.3_{\pm 3.32}$ / $61.3_{\pm 3.58}$ | $64.9_{\pm 3.88}$ / $63.8_{\pm 6.12}$ / $62.6_{\pm 2.89}$ | $54.8_{\pm 1.74}$ / $55.3_{\pm 0.65}$ / $53.7_{\pm 1.09}$ | $59.9_{\pm 2.87}$ / $58.5_{\pm 1.39}$ / $57.8_{\pm 0.84}$ | $35.2_{\pm 0.55}$ / $36.3_{\pm 1.79}$ / $35.7_{\pm 1.20}$ | $41.7_{\pm 0.34}$ / $41.4_{\pm 0.39}$ / $40.2_{\pm 0.10}$ |
| DistillMatch | $57.8_{\pm 6.45}$ | $59.4_{\pm 1.67}$ | $35.7_{\pm 1.78}$ | $41.3_{\pm 1.96}$ | $21.8_{\pm 0.49}$ | $26.2_{\pm 2.05}$ |
| OpenACL | $64.3_{\pm 2.75}$ | $66.3_{\pm 1.17}$ | $60.4_{\pm 1.19}$ | $66.6_{\pm 0.28}$ | $40.2_{\pm 0.45}$ | $47.0_{\pm 0.42}$ |

### A.4 ADDITIONAL EXPERIMENTS

Open SSCL shares some common settings as Novel Class Discovery problems (NCD) (Joseph et al., 2022; Roy et al., 2022). NCD considers unlabeled data only has novel classes, so $C_u \cap C_l = \emptyset$. In the continual learning setting, class-incremental NCD (Roy et al., 2022) leverages the model pre-trained on sequential labeled data to discover novel categories in an unlabeled data set. Usually, NCD methods (Joseph et al., 2022; Roy et al., 2022; Han et al., 2020; Wang et al., 2020) have separate training phases: training on labeled datasets, and discovering novel classes in the unlabeled dataset. Compared with the NCD problem, Open SSCL operates under the assumption that unlabeled data comprises both known and unknown classes, thus defined as $C_u = C_l \cup C_n$. In addition, Open SSCL concurrently learns from labeled and unlabeled datasets and uses the information from novel classes to enhance the model's adaptability for future tasks.

Table 7: Table 2 with standard deviation

| Method | CIFAR-100 | | Tiny-ImageNet | |
|---|---|---|---|---|
| Labels % | 20 | 50 | 20 | 50 |
| Single | $3.1_{\pm0.20}$ / $2.8_{\pm0.20}$ / $2.5_{\pm0.09}$ | $3.0_{\pm0.37}$ / $2.5_{\pm0.69}$ / $3.0_{\pm0.31}$ | $1.9_{\pm0.09}$ / $2.0_{\pm0.11}$ / $1.7_{\pm0.12}$ | $2.4_{\pm0.12}$ / $2.8_{\pm0.27}$ / $2.7_{\pm0.13}$ |
| iCaRL | $6.8_{\pm1.19}$ / $7.0_{\pm0.56}$ / $6.3_{\pm1.25}$ | $7.3_{\pm0.66}$ / $8.3_{\pm0.50}$ / $7.0_{\pm0.96}$ | $4.5_{\pm0.95}$ / $3.3_{\pm0.19}$ / $3.4_{\pm0.30}$ | $4.1_{\pm0.29}$ / $4.8_{\pm0.36}$ / $4.2_{\pm0.31}$ |
| DER | $3.7_{\pm0.11}$ / $3.7_{\pm0.23}$ / $3.5_{\pm0.31}$ | $3.6_{\pm0.23}$ / $3.9_{\pm0.57}$ / $3.9_{\pm0.81}$ | $2.4_{\pm0.11}$ / $2.5_{\pm0.13}$ / $2.1_{\pm0.19}$ | $2.4_{\pm0.10}$ / $2.6_{\pm0.16}$ / $2.3_{\pm0.27}$ |
| GEM | $7.0_{\pm0.14}$ / $8.0_{\pm0.47}$ / $6.9_{\pm1.48}$ | $9.7_{\pm1.06}$ / $7.7_{\pm2.15}$ / $6.7_{\pm2.27}$ | $2.4_{\pm0.08}$ / $3.4_{\pm0.24}$ / $2.7_{\pm0.17}$ | $2.3_{\pm0.66}$ / $2.6_{\pm0.09}$ / $1.8_{\pm0.44}$ |
| GSS | $12.8_{\pm0.64}$ / $11.2_{\pm0.32}$ / $10.3_{\pm1.28}$ | $16.8_{\pm1.11}$ / $15.3_{\pm2.27}$ / $15.2_{\pm1.54}$ | $3.3_{\pm0.21}$ / $5.4_{\pm0.63}$ / $3.8_{\pm0.33}$ | $5.3_{\pm0.40}$ / $5.6_{\pm0.36}$ / $5.0_{\pm0.13}$ |
| ER | $10.9_{\pm0.71}$ / $12.0_{\pm0.84}$ / $11.5_{\pm1.38}$ | $15.6_{\pm0.93}$ / $15.8_{\pm0.98}$ / $16.9_{\pm0.45}$ | $3.3_{\pm0.06}$ / $4.2_{\pm0.46}$ / $3.9_{\pm0.15}$ | $4.8_{\pm0.22}$ / $6.7_{\pm0.61}$ / $5.7_{\pm0.31}$ |
| DistillMatch | $2.8_{\pm0.06}$ | $3.2_{\pm0.17}$ | $2.0_{\pm0.18}$ | $2.7_{\pm0.14}$ |
| **OpenACL** | $\mathbf{15.7}_{\pm0.44}$ | $\mathbf{20.0}_{\pm1.23}$ | $\mathbf{7.9}_{\pm0.37}$ | $\mathbf{11.9}_{\pm1.06}$ |

In this section, we conducted a comparative analysis of our approach with existing NCD methods. We adapt them to the Open SSCL framework to make them simultaneously train on both labeled and unlabeled datasets. For baseline methods that require a pre-training phase, we utilized SimCLR to pre-train the models. Our comparison encompassed the following methods:

1. *AutoNovel* (Han et al., 2020): AutoNovel is designed for the NCD problem by first training on the labeled dataset and then transferring to the unlabeled dataset to discover novel classes using rank statistics.

2. *FRoST* (Roy et al., 2022): FRoST uses feature replay and knowledge distillation on labeled data to prevent forgetting and then use pseudo-labeling to discover novel classes in the unlabeled dataset for class-incremental NCD.

3. *FACT* (Zhou et al., 2022): FACT reserves the embedding space for future tasks.

Also, we further compare some methods specifically designed for online continual learning (OCL):

1. *ER-ACE* (Caccia et al., 2022): It deploys asymmetric cross-entropy for online continual learning.

2. *DVC* (Huo et al., 2023): DVC improves representations with contrastive learning for online continual learning. We extend their contrastive learning module to our setting.

In our study, we present the average accuracy across three runs on both Task-IL and Class-IL benchmarks, as detailed in Tables 8 and 10. These results demonstrate that NCD methods outperform general continual learning approaches in Table 8 and 10. However, it is noteworthy that OpenACL exhibits even stronger performance than NCD methods, underscoring the effectiveness of our proposed method. The comparison with OCL baselines further proves the advantages of employing OpenACL in addressing the Open SSCL problem. Additionally, we also report the BWT and FWT on Task-IL benchmarks in Table 9. OpenACL still achieves the best FWT among these baselines-demonstrating its superior zero-shot learning ability.

Table 8: Average accuracy over three runs of experiments on Task-IL benchmarks.

| Method | CIFAR-10 | | CIFAR-100 | | Tiny-ImageNet | |
|---|---|---|---|---|---|---|
| Labels % | 20 | 50 | 20 | 50 | 20 | 50 |
| AutoNovel | $56.3_{\pm1.82}$ | $56.5_{\pm2.11}$ | $58.7_{\pm0.13}$ | $63.3_{\pm0.83}$ | $37.4_{\pm0.74}$ | $43.1_{\pm4.74}$ |
| FRoST | $54.2_{\pm1.99}$ | $54.9_{\pm1.36}$ | $53.1_{\pm0.60}$ | $57.9_{\pm0.84}$ | $33.0_{\pm0.96}$ | $41.1_{\pm1.31}$ |
| FACT | $53.2_{\pm3.27}$ | $55.3_{\pm1.78}$ | $55.9_{\pm2.86}$ | $62.8_{\pm1.00}$ | $35.0_{\pm1.49}$ | $42.3_{\pm0.67}$ |
| ER-ACE | $61.2_{\pm1.83}$ / $61.6_{\pm3.78}$ / $61.3_{\pm2.45}$ | $62.4_{\pm0.91}$ / $64.2_{\pm2.95}$ / $63.9_{\pm1.99}$ | $53.8_{\pm2.08}$ / $55.0_{\pm0.78}$ / $54.8_{\pm1.78}$ | $61.7_{\pm0.71}$ / $62.4_{\pm0.93}$ / $62.1_{\pm0.86}$ | $36.2_{\pm1.36}$ / $37.2_{\pm0.78}$ / $35.4_{\pm1.25}$ | $41.4_{\pm0.54}$ / $42.4_{\pm1.63}$ / $40.6_{\pm0.74}$ |
| DVC | $57.4_{\pm0.86}$ | $61.7_{\pm3.23}$ | $57.6_{\pm0.92}$ | $62.7_{\pm2.08}$ | $36.8_{\pm0.61}$ | $43.5_{\pm0.35}$ |
| **OpenACL** | $\mathbf{64.3}_{\pm2.75}$ | $\mathbf{66.3}_{\pm1.17}$ | $\mathbf{60.4}_{\pm1.19}$ | $\mathbf{66.6}_{\pm0.28}$ | $\mathbf{40.2}_{\pm0.45}$ | $\mathbf{47.0}_{\pm0.42}$ |

Table 9: BWT and FWT results on 50% labeled dataset. The results show as BWT / FWT.

|  | AutoNovel | FRoST | FACT | ER-ACE | DVC | OpenACL |
|---|---|---|---|---|---|---|
| CIFAR-100 | 10.6 / 1.1 | 5.7 / 1.4 | 7.8 / 2.4 | **12.4** / -1.7 | 11.1 / 1.6 | 9.2 / **13.0** |
| Tiny-ImageNet | 4.6 / 0.9 | -0.1 / 0.7 | 3.7 / 3.9 | **6.0** / -0.1 | 5.9 / 0.5 | 2.7 / **10.9** |

Table 10: Average accuracy over three runs of experiments on Class-IL benchmarks.

| Method | CIFAR-100 | | Tiny-ImageNet | |
|---|---|---|---|---|
| Labels % | 20 | 50 | 20 | 50 |
| AutoNovel | $13.2_{\pm 0.61}$ | $17.9_{\pm 1.19}$ | $6.5_{\pm 0.57}$ | $9.2_{\pm 0.58}$ |
| FRoST | $7.6_{\pm 0.46}$ | $10.5_{\pm 0.84}$ | $3.7_{\pm 0.09}$ | $4.3_{\pm 0.18}$ |
| FACT | $12.9_{\pm 0.84}$ | $16.3_{\pm 0.89}$ | $5.9_{\pm 0.90}$ | $8.2_{\pm 1.18}$ |
| ER-ACE | $12.8_{\pm 0.20}$ / $13.3_{\pm 0.90}$ / $12.0_{\pm 0.79}$ | $16.7_{\pm 0.79}$ / $17.9_{\pm 0.63}$ / $17.1_{\pm 1.20}$ | $5.0_{\pm 0.55}$ / $5.4_{\pm 0.56}$ / $4.9_{\pm 0.36}$ | $7.4_{\pm 0.74}$ / $8.1_{\pm 0.90}$ / $7.2_{\pm 0.52}$ |
| DVC | $11.2_{\pm 0.78}$ | $16.2_{\pm 2.08}$ | $5.8_{\pm 0.40}$ | $8.3_{\pm 1.42}$ |
| **OpenACL** | $\mathbf{15.7}_{\pm 0.44}$ | $\mathbf{20.0}_{\pm 1.23}$ | $\mathbf{7.9}_{\pm 0.37}$ | $\mathbf{11.9}_{\pm 1.06}$ |

