# OpenReview forum: "Exploiting Open-World Data for Adaptive Continual Learning"
_ICLR.cc/2024/Conference — Submitted to ICLR 2024_

### Official Review · Reviewer_58yd · 2023-10-25

**Soundness:** 2 fair
**Presentation:** 3 good
**Contribution:** 2 fair
**Rating:** 5
**Confidence:** 3

**Summary:**

This paper studies continual learning in open-set scenarios and proposes a prototype-based approach, OpenACL, to leverage open-world data to enhance model performance on new tasks while mitigating catastrophic forgetting. The authors also demonstrate the effectiveness of the proposed method in online continual learning setting through experiments on CIFAR-10, CIFAR-100, and Tiny-ImageNet.

**Strengths:**

- To the best of my knowledge, this paper is the first to combine open semi-supervised learning with continual learning, so the work is novel.
- The motivation to improve the model's generalization ability to new tasks by leveraging unlabeled OOD data rather than eliminating it directly is reasonable.
- The paper is well written and easy to follow.

**Weaknesses:**

1. My biggest concern is that the problem definition of Open SSCL seems unreasonable. The authors assume that large amounts of unlabeled data can be accessed at any time and are not affected by task switching. Furthermore, the input x for unlabeled and labeled samples comes from the same distribution. Such a setting is incompatible with traditional continual learning to a certain extent, or reduces the difficulty of its core problem, catastrophic forgetting. In particular, the ablation experiments provided by the appendix also show that without unlabeled data, the proposed method is inferior to ER in terms of accuracy.
It is recommended that the authors provide more examples or explanations on the practicality and importance of the problem definition, rather than simply combining Open SSL and CL settings.


2. The experiments only focus on the setting of online continual learning (OCL), but the selected baselines are not specifically designed for OCL. Could the authors compare the results of offline continual learning? Or Could it be compared with the OCL methods mentioned in [1]?

3. The authors mention that the hyperparameters for baselines are set to the suggested value in their original implementation, but when the experimental settings are different from the original paper, how can the authors ensure that the setting of the hyperparameters is fair?

[1] Wang L, Zhang X, Su H, et al. A comprehensive survey of continual learning: Theory, method and application[J]. arXiv preprint arXiv:2302.00487, 2023.

**Questions:**

1. In Table 1, why are the results worse for Independent using SimCLR/FixMatch?  Could the authors provide further analysis?
2. The baseline Single denotes training a single network on data from all tasks. Why is it affected by the new task in Figure 2 & 3? (It is recommended that the legend not block the image content)
3. Although the paper focuses on OCL settings, it seems to be limited to labeled datasets only. Because unlabeled data can be accessed at any time, as more and more tasks are learned, unlabeled data is learned more and more times. Have the authors considered that the unlabeled dataset will also change as the task changes?

---

> ### Author Response · Authors · 2023-11-18
> **To Review 58yd**
>
> We appreciate the helpful and concrete suggestions. We address your concerns below.
>
> ### RE: Concern of the problem definition
>
> In real applications, especially those involving continual learning, obtaining a steady stream of labeled data can be difficult. This is particularly hard for new or rapidly evolving domains. Continually labeling large datasets can be very expensive and time-consuming, requiring significant human effort, especially for complex tasks or specialized domains. However, it is not as expensive and difficult as labeled data to obtain large amounts of unlabeled data. More importantly, the acquired unlabeled data in practice often come from different (possibly never-before-seen) classes. Therefore, we believe Open SSCL is reasonable and well-motivated by real applications. Also, we argue that these unlabeled data can be effectively utilized to improve model adaptability on future tasks while tackling the catastrophic forgetting issue in continual learning.
>
> Regarding your concerns about the assumption imposed on unlabeled data. We want to clarify that our method does **not** have any restriction on unlabeled data, i.e., we allow unlabeled data to change as task switches and the unlabeled data actually comes from **different** distributions of labeled data. In our experiments, each unlabeled sample is only used **once** throughout the learning process, so the unlabeled dataset used for each task is different. Because unlabeled data includes data from all classes (including known classes, novel classes in future tasks, and OOD classes that never appear in continual learning tasks) while the labeled data do not contain novel/OOD classes, the input distributions for labeled and unlabeled samples can be significantly different.
>
> ### RE: without unlabeled data, the proposed method is inferior to ER
>
> Without unlabeled data, our proposed method (OpenACL) just uses Eq.(2) to optimize the model, and the problem is reduced to conventional continual learning. In this case, OpenACL and ER can attain similar results. The difference is that ER uses the linear transformation to compute the probability for each class while OpenACL uses the cosine similarity to the prototypes to get the probability. However, note that the key novelty of our work is to leverage unlabeled data to improve model adaptability on future tasks while mitigating catastrophic forgetting in continual learning. In the presence of unlabeled data, OpenACL consistently outperforms other algorithms across various datasets.
>
> ### RE: Provide more examples or explanations on the practicality and importance of the problem definition
>
> Thanks for your suggestion, we have updated a new version and added more explanations in the introduction section.

---

> > ### Author Response · Authors · 2023-11-18
> >
> > ### RE:Compare with the OCL methods mentioned in [1]
> >
> > We compare OpenACL with two methods for OCL mentioned in [1]: ER-ACE [A] and DVC [B]. ER-ACE is also augmented by SimCLR and FixMatch as other baselines in our experiments. For DVC, we extend their contrastive learning module to our setting. We show the results below:
> >
> > | **Method/Task-IL** | **CIFAR-10 20%**               | **CIFAR-10 50%**               | **CIFAR-100 20%**              | **CIFAR-100 50%**              | **Tiny-ImageNet 20%**          | **Tiny-ImageNet 50%**          |
> > | ------------------------ | ------------------------------------ | ------------------------------------ | ------------------------------------ | ------------------------------------ | ------------------------------------ | ------------------------------------ |
> > | ER-ACE                   | 61.2±1.83 / 61.6±3.78 / 61.3±2.45 | 62.4±0.91 / 64.2±2.95 / 63.9±1.99 | 53.8±2.08 / 55.0±0.78 / 54.8±1.78 | 61.7±0.71 / 62.4±0.93 / 62.1±0.86 | 36.2±1.36 / 37.2±0.78 / 35.4±1.25 | 41.4±0.54 / 42.4±1.63 / 40.6±0.74 |
> > | DVC                      | 57.4±0.86                           | 61.7±3.23                           | 57.6±0.92                           | 62.7±2.08                           | 36.8±0.61                           | 43.5±0.35                           |
> > | **OpenACL**        | **64.3±2.75**                      | **66.3±1.17**                      | **60.4±1.19**                      | **66.6±0.28**                      | **40.2±0.45**                      | **47.0±0.42**                      |
> >
> > | **Method/Class-IL** | **CIFAR-100 20%**              | **CIFAR-100 50%**              | **Tiny-ImageNet 20%**       | **Tiny-ImageNet 50%**       |
> > | ------------------------- | ------------------------------------ | ------------------------------------ | --------------------------------- | --------------------------------- |
> > | ER-ACE                    | 12.8±0.20 / 13.3±0.90 / 12.0±0.79 | 16.7±0.79 / 17.9±0.63 / 17.1±1.20 | 5.0±0.55 / 5.4±0.56 / 4.9±0.36 | 7.4±0.74 / 8.1±0.90 / 7.2±0.52 |
> > | DVC                       | 11.2±0.78                           | 16.2±2.08                           | 5.8±0.40                         | 8.3±1.42                         |
> > | **OpenACL**         | **15.7±0.44**                 | **20.0±1.23**                 | **7.9±0.37**               | **11.9±1.06**              |
> >
> > The results on Task-IL and Class-IL benchmarks show that OpenACL still outperforms methods that are specifically designed for OCL.
> >
> > ### RE: Hyperparameter choice
> >
> > Indeed, we have tried different combinations of hyper-parameters in our experiments and found that the suggested values in the original implementation for each dataset are still optimal in our setting. We have clarified this in the revised version.
> >
> > ### RE: why are the results worse for Independent using SimCLR/FixMatch?
> >
> > The *Independent* baseline applies separate networks for each task. Therefore, in each task, the network is trained from sketch and is sensitive to the number of labeled data. We observe that *Independent* with SimCLR/FixMatch is worse than only using supervised learning when the labeled data is limited (e.g., when the ratio of labeled data is 20\% on CIFAR-100 and Tiny-ImageNet in Table 1 of the manuscript.). Because each network is trained from sketch, the *Independent* method does not have enough ground truth to learn from and the insufficient labeled data can lead to the poor quality of pseudo-labels for FixMatch. Similarly, SimCLR relies on the quality of representations learned from the data which makes it limited to the method with multiple separate networks.
> >
> > ### RE: The baseline Single denotes training a single network on data from all tasks; Figure Legend
> >
> > We have updated section 5.1 of the manuscript to make *Single* clearer. *Single* is trained across all tasks. It still sequentially trains on each task, not from all data in one task. So it is affected by new tasks.
> >
> > Thanks for your suggestion for the legend in the figures. We have corrected it.
> >
> > ### RE: Have the authors considered that the unlabeled dataset will also change as the task changes?
> >
> > We want to clarify that the unlabeled datasets used for each task are indeed different in our method. In experiments, each unlabeled data is only used **once** so that the unlabeled dataset changes as the task changes. It is equivalent to randomly dividing the unlabeled dataset and assigning each portion to each task.
> >
> > [A] Caccia, Lucas, Rahaf Aljundi, Nader Asadi, Tinne Tuytelaars, Joelle Pineau, and Eugene Belilovsky. "New Insights on Reducing Abrupt Representation Change in Online Continual Learning." In International Conference on Learning Representations. 2021.
> >
> > [B] Huo, Fushuo, Wenchao Xu, Jingcai Guo, Haozhao Wang, Yunfeng Fan, and Song Guo. "Offline-online class-incremental continual learning via dual-prototype self-augment and refinement." arXiv preprint arXiv:2303.10891 (2023).

---

> > ### Comment · Reviewer_58yd · 2023-11-21
> >
> > Thanks for the rebuttal. The new experiments can further illustrate the effectiveness of the proposed method for the problem Open SSCL raised by the authors. However, I think there are still some unclear points in the definition of the Open SSCL problem and the proposed method, as follows:
> >
> > 1. The authors mentioned that "continually labeling large datasets can be very expensive and time-consuming, requiring significant human effort, especially for complex tasks or specialized domains", I think this is the core problem that Open SSL wants to solve. For Open SSCL, in addition to the above issues, we also need to pay attention to the catastrophic forgetting caused by changes in the data flow (including labeled and unlabeled). Although the authors emphasize that they have no restrictions on unlabeled samples, such a setting is too vague and does not allow for further analysis of the degree of forgetting. Moreover, the authors did not provide some actual scenarios to support the practicality and rationality of their settings.
> >
> > 2. The authors mentioned that "each unlabeled sample is only used once throughout the learning process". Does this mean that each unlabeled sample will only be used once during the learning process of all tasks? If so, how many unlabeled samples will be used in the learning process of each task? How do you select samples? What to do if the task has not been learned yet, but all unlabeled samples have been used? Will the algorithm's effectiveness be affected if different strategies are used for the above questions?
> >
> > Based on the above considerations, I am currently maintaining my score.

---

> > > ### Author Response · Authors · 2023-11-21
> > >
> > > ### RE: Question 1
> > >
> > > **More clarification on unlabeled data in Open SSCL.** In practical applications, unlabeled data is usually collected from the web or from the user data (if we are allowed to use user data as training data) whose qualify is out of our control. Therefore, we cannot specify the time-varying pattern of the unlabeled dataset. In our experiment, we construct our unlabeled dataset as a combination of previous tasks, current tasks, future tasks, and OOD data, and only use each unlabeled data once to make it change across tasks.
> > >
> > > Although we don't have any restriction on unlabeled data, we can still assess the degree of forgetting on previous tasks by applying the Backward Transfer (BWT) metric, as demonstrated in Table 3. Note that in the context of semi-supervised learning, the primary role of the unlabeled dataset is to improve performance on labeled tasks. Therefore, we focus on evaluating the forgetting of continual tasks instead of unlabeled datasets.
> > >
> > > **Actual scenarios of our setting.** One example is an industrial scenario considering the rapid deployment of an image classification system. After launching the initial model, we observed a significant number of user-uploaded images that didn't fit into predefined classes. To fit these novel/OOD classes, the model requires a continuous collection of sample-label pairs, making it increasingly challenging due to the continuously expanding variety of possible classes. However, unlabeled datasets cover a large number of data with various classes. By leveraging unlabeled datasets, we can efficiently cluster OOD data with similar features, preparing it for incorporation into future classes. When we need to train the new classes (new tasks), this strategy reduces the effort required to generate adequate sample-label pairs and speeds up the model training process. Open SSCL considers such a problem, so we propose to continually mine the hidden pattern in unlabeled datasets to adapt the model to new tasks.
> > >
> > > Similarly, Open SSCL can be adapted to scenarios such as social media platforms that regularly update content moderation rules, or banking systems evolving to detect new fraud techniques. By uncovering hidden patterns in unlabeled datasets, this approach enables quicker adaptation to new tasks.
> > >
> > > ### RE: Question 2
> > >
> > > Yes, each unlabeled sample is only used once during the learning process of all tasks. We show how we split the labeled part and unlabeled part in our dataset explanation (Section 5.1). For example, in CIFAR-100 (20\%), 20\% data are labeled for each task class. So, for the 80 task classes (40,000 instances), we have 8,000 labeled instances and 32,000 unlabeled instances. These 8,000 instances are split into 16 tasks. Then, we have 20 OOD classes (10,000 instances) in the unlabeled dataset, so we have 42,000 unlabeled instances, consisting of 32000 instances from task classes and 10000 instances from OOD classes. In this case, each task has 500 labeled instances, so we access 2,625 unlabeled instances in each task to use all data after 16 tasks. We have also updated the manuscript to include more details of the dataset split in Appendix A.1.3.
> > >
> > > To ensure that each unlabeled sample is used only once, we adjust the ratio of labeled to unlabeled data in each training batch to match the true proportion in the datasets.  So, after using all labeled data, every unlabeled data is visited.
> > >
> > > Note that, this setting is used to align the usage of the unlabeled dataset with the principles of online continual learning, where different sets of data are used for different tasks.  If all unlabeled samples have been used, it becomes feasible to reuse them. In semi-supervised learning, it is common to reuse unlabeled data more than once [A,B].
> > >
> > > To evaluate the effectiveness, we conducted comparisons using varying proportions of unlabeled data: 80\% and 50\% of the data being unlabeled. In future work, we plan to explore the impact of reusing unlabeled data multiple times in an online continual learning framework. It's anticipated that training on unlabeled datasets more than once could lead to more efficient utilization, as it may learn the better representation and enhance the grouping of samples under their respective prototypes after repeated training.
> > >
> > > [A] Sohn, Kihyuk, David Berthelot, Nicholas Carlini, Zizhao Zhang, Han Zhang, Colin A. Raffel, Ekin Dogus Cubuk, Alexey Kurakin, and Chun-Liang Li. "Fixmatch: Simplifying semi-supervised learning with consistency and confidence." Advances in neural information processing systems 33 (2020): 596-608.
> > >
> > > [B] Smith, James, Jonathan Balloch, Yen-Chang Hsu, and Zsolt Kira. "Memory-efficient semi-supervised continual learning: The world is its own replay buffer." In 2021 International Joint Conference on Neural Networks (IJCNN), pp. 1-8. IEEE, 2021.

---

> > > > ### Author Response · Authors · 2023-11-21
> > > >
> > > > We appreciate your valuable feedback and hope that our responses address the concerns raised.

---

> > > > ### Comment · Reviewer_58yd · 2023-11-22
> > > >
> > > > Thanks to the authors for their detailed responses.
> > > >
> > > > For response 2, can I understand that the 2,625 unlabeled instances accessed when learning for each task include all 20 OOD classes? In other words, is at least one sample of the 20 OOD classes be accessed during each task learning? If so, I'm wondering if the prototype adaptation module can handle the case of unlabeled classes change. For example, the unlabeled samples accessed when learning different tasks are from different classes.
> > > >
> > > > I strongly recommend that the authors describe more clearly how to sample unlabeled data in Section 5.1.

---

> > > > > ### Author Response · Authors · 2023-11-22
> > > > >
> > > > > Thank you for your important questions and recommendations.
> > > > >
> > > > > Regarding response 2, yes, the 2,625 unlabeled instances **include all 20 OOD classes** as well as other the 80 classes for all tasks. The unlabeled dataset is randomly shuffled. Thus, for each task, we can access all classes in the unlabeled dataset.
> > > > >
> > > > > When learning different tasks, the set of unlabeled classes still includes all classes in continual tasks and OOD classes. For each task, we consider the unlabeled instances are different, but they come from the same classes (e.g. 100 classes in CIFAR-100). Therefore, the **unlabeled classes don't change**; only the labeled samples are from different classes for different tasks.
> > > > >
> > > > > The prototype adaptation module aims to find some closest prototypes to the new classes in a new task. These classes should be present in unlabeled datasets, but their labels are unknown until we shift to the new task. The prototype adaptation module **handles the labeled classes** change and assigns most likely prototypes to the new classes in the new task, enabling our model to adapt to the new task.
> > > > >
> > > > > Assuming that unlabeled classes change when shifting task in your case, it is anticipated that the prototype adaptation module will still be capable of identifying the most likely prototypes for the classes in the new task if these classes have been presented in previous training. Since we employ K-means to reinitialize prototypes for novel classes, it is feasible to learn prototypes for new unlabeled classes in the new task in this scenario. In future work, we plan to further explore how to leverage the dynamics of changing unlabeled classes.
> > > > >
> > > > > Thank you for your valuable suggestion regarding Section 5.1. We have revised this section to include more details about the sampling of unlabeled data. Additionally, to illustrate our approach more clearly, we have included a specific example demonstrating how we sample unlabeled data in the last paragraph of Appendix A.1.3.

---

> > > > > > ### Author Response · Authors · 2023-11-23
> > > > > >
> > > > > > Thank you very much for taking the time and effort to review our paper. We kindly remind you that the discussion period between reviewers and authors is approaching its final hours. We hope that our responses and the revised manuscript adequately address all the concerns. We would be grateful if you could confirm whether your concerns have been adequately addressed, or if you have any further comments. We sincerely appreciate your valuable feedback and guidance.

---

### Official Review · Reviewer_rvMJ · 2023-10-26

**Soundness:** 2 fair
**Presentation:** 3 good
**Contribution:** 2 fair
**Rating:** 6
**Confidence:** 4

**Summary:**

The paper is concerned with open-world continual learning. The proposed approach (called OpenACL) is based on the assumption that the novel classes in the out-of-distribution (OOD) data for the current task may become training data in future tasks. Instead of identifying and rejecting OOD samples, the authors use them to adapt a model to a new task and improve the model performance in CL. The proposed approach maintains multiple prototypes for seen tasks and reserves extra prototypes for unseen tasks. Both labeled and unlabeled
data are learned to improve the adaptation ability and tackle catastrophic forgetting for prototypes.

**Strengths:**

The paper is, in general, clearly written, well-documented and easy to follow. The review of the state of the art covers almost the relevant literature. The experimental results are extensive and demonstrate the superiority of the proposed approach.

**Weaknesses:**

There is a confusion of the concepts and terminology being used (see the Questions section)

**Questions:**

The authors need to clarify the following aspects:
- They claim (first paragraph, page 2, middle): "Instead of identifying and eliminating OOD samples during training, we may leverage...". The problem of OOD detection is addressed at inference time, not during training. You could simply refer as: unlabeled training data or semi-supervised learning. Please correct this aspect through the paper and clarify which problem you want to address.
- Since you want to identify and label unknown samples, your problem is related to Novel Class Discovery problem, not OOD detection.
See for instance references [1, 2] (below). Please discuss and clarify these aspects in the paper and update accordingly the related work section.
- The idea of reserving new prototypes for future classes is not new. It was addressed in [3] (see below). However, it limits considerably the generalization capability of your approach. In a real-world problem, you do not know how the distribution of novel classes will look like, so it is not posible to 'pre-define' prototypes. Some other approaches such as self-supervised learning might be preferred instead to discover new clusters in the unlabeled data.
- Compare your approach with some methods for Novel Class Discovery


References:
[1] Subhankar Roy, Mingxuan Liu, Zhun Zhong, Nicu Sebe, Elisa Ricci. Class-incremental Novel Class Discovery (ECCV 2022)
[2] K J Joseph, Sujoy Paul, Gaurav Aggarwal, Soma Biswas, Piyush Rai, Kai Han, Vineeth N Balasubramanian. Novel Class Discovery without Forgetting (ECCV 2022)
[3] Da-Wei Zhou, Fu-Yun Wang, Han-Jia Ye, Liang Ma, Shiliang Pu, De-Chuan Zhan. Forward Compatible Few-Shot Class-Incremental Learning. (CVPR 2022)

---

> ### Author Response · Authors · 2023-11-18
> **To Review rvMJ**
>
> We thank the reviewer for the constructive review and suggestions. We address each of your concerns in the subsequent response.
>
> ### RE: The problem of OOD detection is addressed at inference time, not during training.
>
> We acknowledge the problem of classic OOD detection is addressed at inference time, not during training. However, in a **semi-supervised** open-set problem, it is necessary to consider OOD data during training because unseen OOD data in the unlabeled dataset would harm model training [A]. The problem of *Open-Set/World Semi-Supervised Learning* has been studied in the literature and we have discussed it in the related work section. For example, [A] assigns lower weights for outliers during training to keep the model safety. [B] adds an additional OOD filtering process into the semi-supervised object detection training pipeline to remove OOD instances.
>
> In this work, we extend the problem to continual learning (CL) and formulate our problem as open semi-supervised continual learning. Similar to [A,B], we also use unlabeled OOD data during training but the main novelty is to exploit these unlabeled OOD data to improve model adaptability on future tasks while simultaneously tackling catastrophic forgetting in CL by using unlabeled ID data.
>
> ### RE: Relationship to Novel Class Discovery Problem
>
> Our work is motivated by [C] which also studies the open-world semi-supervised learning problem. Unlike classic novel class discovery that only considers novel classes in unlabeled datasets and often involves a separate training procedure (i.e., it only discovers novel classes \textit{after} training on labeled datasets), open-world SSL includes both seen classes and unseen classes in unlabeled datasets and does not require separate training.
>
> In this paper, we extend the open-world semi-supervised learning problem to the continual learning setting and our key novelty is to continually leverage novel classes (OOD data) to enhance the model adaptability during one-time training. Note that other semi-supervised continual learning works with novel classes (e.g., novel class discovery in [1,2], open-set in [D,E]) don't exploit these data to adapt to the task stream. We will add these works you mentioned to the manuscript.
>
> Although our work is related to novel class discovery, we want to clarify that the fundamental setting of OpenACL is *Open-world SSL*, **not** novel class discovery. Thus, our work is fundamentally different from [1,2]. Specifically, [1] and [2] explicitly specified the classes in labeled and unlabeled datasets, and assume they are disjoint, i.e., $Y_{lab} \cap Y_{unlab}= \emptyset $ . Using the notations in our paper, it would be $C_u \cap C_l= \emptyset$ which is different from our setting where $C_u = C_l \cup  C_n$. Since there is no risk of misclassifying novel classes as known classes when $C_u \cap C_l= \emptyset$, the novel class discovery in [1,2] is easier than open-world setting we considered in this work.
>
> In addition, the novel classes $C_n$ considered in our work consist of both classes in upcoming tasks and OOD classes that never appear in continual tasks. Because the classes of OOD data may be labeled ID classes in upcoming tasks, our method leverages OOD data instead of rejecting them.  Furthermore, under open-world SSL, we can access data for previous tasks so that our prototype learning mechanism can be designed accordingly to mitigate catastrophic forgetting. These significantly differ from novel class discovery.

---

> > ### Author Response · Authors · 2023-11-18
> >
> > ### RE: Prototypes and generalization capability
> >
> > We want to clarify that our method does **not** reserve the classification space for future classes and is different from [3]. Specifically, [3] first trains on the given known classes and reserves pre-allocate space for future classes. After training the given known classes, [3] uses the average embedding of a new class as a prototype during inference. Because pre-allocating space for new classes can limit applicability for real scenarios, our method makes prototypes and data representations automatically approach each other without the need to reserve the space.
> >
> > Moreover, our method does **not** rely on pre-defined prototypes. The prototypes $g$ are determined by trainable parameters and reinitialized by K-means when shifting tasks. Also, we do **not** require prior knowledge of the distribution of novel classes. We only specify the number of prototypes and assume the data from the same class should be close to each other in the latent space, so we employ prototype contrastive learning. The prototype contrastive loss aligns data representations and groups data with similar representations under the same prototype.
> >
> > Ideally, we want the number of prototypes $|g|$ to match the number of all classes $|C_u|$ in the dataset. Even if $|g|<|C_u|$, our method still works and we can tolerate data from different unknown classes going to the same prototype. This is because after assigning the prototype to the class with the most grouped samples during prototype adaptation, Eq.(2) and (4) can still make data from different classes away from this prototype. Note that Eq.(4) is the **self-supervised** prototype contrastive learning loss, and we reinitialized the parameters of $g$ with the centroids of K-means to make prototypes diverse after adaption.
> >
> > As our prototypes are dynamically learned through self-supervised and supervised prototype-level contrastive learning, **OpenACL doesn't limit the generalization capability**. Even under the worst cases where all prototypes are allocated to existing classes and novel classes emerge, we can still initialize some additional prototypes and use them to align the remaining data.
> >
> > ### RE: Compare with some methods for Novel Class Discovery
> >
> > We compare our method with three methods for Novel Class Discovery: AutoNovel [F], FRoST [1], and FACT [3]. We adapt them to the Open SSCL setting and train them using labeled data and unlabeled data simultaneously. We show the results for Task IL benchmarks and Class IL benchmarks below:
> >
> > | **Method/Task-IL** | **CIFAR-10 20%** | **CIFAR-10 50%** | **CIFAR-100 20%** | **CIFAR-100 50%** | **Tiny-ImageNet 20%** | **Tiny-ImageNet 50%** |
> > | ------------------------ | ---------------------- | ---------------------- | ----------------------- | ----------------------- | --------------------------- | --------------------------- |
> > | AutoNovel                | 56.3±1.82             | 56.5±2.11             | 58.7±0.13              | 63.3±0.83              | 37.4±0.74                  | 43.1±4.74                  |
> > | FRoST                    | 54.2±1.99             | 54.9±1.36             | 53.1±0.60              | 57.9±0.84              | 33.0±0.96                  | 41.1±1.31                  |
> > | FACT                     | 53.2±3.27             | 55.3±1.78             | 55.9±2.86              | 62.8±1.00              | 35.0±1.49                  | 42.3±0.67                  |
> > | **OpenACL**        | **64.3±2.75**   | **66.3±1.17**   | **60.4±1.19**    | **66.6±0.28**    | **40.2±0.45**        | **47.0±0.42**        |
> >
> > | **Method/Class-IL** | **CIFAR-100 20%** | **CIFAR-100 50%** | **Tiny-ImageNet 20%** | **Tiny-ImageNet 50%** |
> > | ------------------------- | ----------------------- | ----------------------- | --------------------------- | --------------------------- |
> > | AutoNovel                 | 13.2±0.61              | 17.9±1.19              | 6.5±0.57                   | 9.2±0.58                   |
> > | FRoST                     | 7.6±0.46               | 10.5±0.84              | 3.7±0.09                   | 4.3±0.18                   |
> > | FACT                      | 12.9±0.84              | 16.3±0.89              | 5.9±0.90                   | 8.2±1.18                   |
> > | **OpenACL**         | **15.7±0.44**    | **20.0±1.23**    | **7.9±0.37**         | **11.9±1.06**        |
> >
> > Across both tables, our method consistently outperforms other novel class discovery methods on all datasets. We also compare two new online continual learning baselines that can be found in Appendix A.4 of our latest version.

---

> > > ### Author Response · Authors · 2023-11-18
> > >
> > > [A] Lan-Zhe Guo, Zhen-Yu Zhang, Yuan Jiang, Yu-Feng Li, and Zhi-Hua Zhou. Safe deep semi-supervised learning for unseen-class unlabeled data. In International Conference on Machine Learning, pp. 3897–3906. PMLR, 2020.
> > >
> > > [B] Liu, Y. C., Ma, C. Y., Dai, X., Tian, J., Vajda, P., He, Z., & Kira, Z. (2022, October). Open-set semi-supervised object detection. In European Conference on Computer Vision (pp. 143-159). Cham: Springer Nature Switzerland.
> > >
> > > [C] Kaidi Cao, Maria Brbic, and Jure Leskovec. Open-world semi-supervised learning. In International Conference on Learning Representations (ICLR), 2022.
> > >
> > > [D] Smith, James, Jonathan Balloch, Yen-Chang Hsu, and Zsolt Kira. "Memory-efficient semi-supervised continual learning: The world is its own replay buffer." In 2021 International Joint Conference on Neural Networks (IJCNN), pp. 1-8. IEEE, 2021.
> > >
> > > [E] Wang, Liyuan, Kuo Yang, Chongxuan Li, Lanqing Hong, Zhenguo Li, and Jun Zhu. "Ordisco: Effective and efficient usage of incremental unlabeled data for semi-supervised continual learning." In Proceedings of the IEEE/CVF Conference on Computer Vision and Pattern Recognition, pp. 5383-5392. 2021.
> > >
> > > [F] Han, Kai, Sylvestre-Alvise Rebuffi, Sebastien Ehrhardt, Andrea Vedaldi, and Andrew Zisserman. "Automatically Discovering and Learning New Visual Categories with Ranking Statistics." In International Conference on Learning Representations. 2019.

---

> > > > ### Comment · Reviewer_rvMJ · 2023-11-22
> > > > **Official Comment by Reviewer rvMJ**
> > > >
> > > > After carefully considering authors' responses, I can say that they satisfactorily addressed my concerns. Based on this fact, and taking into account also the other reviewers' comments, I have decided to change my initial rating to: 6. marginally above the acceptance threshold.

---

> > > > > ### Author Response · Authors · 2023-11-22
> > > > >
> > > > > We greatly appreciate your time and valuable feedback! Your comments are essential in helping us improve our work.

---

### Official Review · Reviewer_66AU · 2023-10-29

**Soundness:** 3 good
**Presentation:** 3 good
**Contribution:** 3 good
**Rating:** 6
**Confidence:** 3

**Summary:**

This paper proposes the Open SSCL problem, which aims to leverage unlabeled out-of-distribution (OOD) data to assist continual learning, because these data may become in-distribution (ID) at a future task. To tackle this problem, an algorithm, namely Open ACL is discussed. Specifically, Open ACL consists of three steps. First, the cosine distances between data in seen classes and their prototypes are minimized, to improve the perceived similarity on data points in the same seen class. Second, dissimilarity among data in different classes is improved by contrastive learning. Third, novel prototypes are adapted from centroids of K-means. Experiments show that Open ACL is able to defeat baselines in standard image classification SSCL benchmarks.

**Strengths:**

1. This paper has precisely identified the problem of data shortage in open world continual learning, and proposes to improve learning with additional OOD data. It bridges a gap in state-of-the-art class-incremental continual learning, where OOD data are generally excluded and only ID data is used.
2. The proposed method, Open ACL aims to group data points in the same class with similar representations, while separating data points in different classes with dissimilar ones. This technique is able to handle OOD data from unseen classes, preparing knowledge for encountering them in future.
3. The presentation is easy to follow.

**Weaknesses:**

The experiment design seems to be inconsistent with the method. How are unlabeled OOD data being input per task? I assume that for each task, the model will receive some ID data from its current classes, and some OOD data from future classes. However, it seems that all training data points, including labeled and unlabeled, are ID.

**Questions:**

Please respond to the weakness I mentioned above. If this can be clearly addressed, I am willing to improve my rating.

---

> ### Author Response · Authors · 2023-11-18
> **To Review 66AU**
>
> We thank the reviewer for the positive evaluation of our work. Below is our response to the weakness you mentioned.
>
> Our method takes two datasets as input: labeled $D_l = \{\mathcal{D}_l^1,...,\mathcal{D}_l^k \}$ and unlabeled $D_u$. For each task $i$, we simultaneously sample data $\mathcal{D}_l^i$ from the labeled dataset for the current task $i$ and the unlabeled data $D_u$. Note that $D_u$ consists of data from all classes, including previous task classes, current task classes, future task classes (we haven't seen their labels, so they are considered as OOD for this task), and some pure OOD classes that never appear in continual learning tasks. Thus, OOD data for each task $i$ includes two parts: future tasks classes and OOD classes that never appear in continual tasks. During the training, we sampled the data from the unlabeled dataset without knowing the source, i.e., the data comes from previous task classes, current task classes, future task classes, and pure OOD classes. In other words, the training samples are actually the combination of ID and OOD data.

---

> > ### Comment · Reviewer_66AU · 2023-11-21
> >
> > Thank you for your response. I believe this is a very important experiment setup detail, which should be included in the main text (e.g. Section 5.1), so that the readers can find it with ease. If you can update this in your revised draft, I would be happy to raise my rating.

---

> > > ### Author Response · Authors · 2023-11-21
> > >
> > > Thank you for your valuable feedback. We agree that these experimental setup details are important for understanding the research. Based on your suggestion, we have incorporated these details into Section 5.1 of the main text. We also give a more detailed illustration of datasets in Appendix A.1.3. We appreciate your constructive suggestion and are confident that these changes will improve the quality of the work.

---

> > > > ### Comment · Reviewer_66AU · 2023-11-21
> > > >
> > > > Thank you for the update. I have changed my ratings accordingly.

---

> > > > > ### Author Response · Authors · 2023-11-21
> > > > >
> > > > > We sincerely thank you for your valuable time and precious rating. Your constructive comments have helped to improve this work a lot.

---

### Official Review · Reviewer_9HZN · 2023-11-05

**Soundness:** 2 fair
**Presentation:** 2 fair
**Contribution:** 2 fair
**Rating:** 5
**Confidence:** 3

**Summary:**

This paper proposes openACL, the open-world continual learning. The author consider the situation that the training data in one task may come from the open-world dataset (OOD), and considers such OOD data (from novel classes) to mine the underlying pattern in unlabeled open-world data. So that the model’s adaptability to upcoming tasks will be  empowered. At last, the author organizes extensive experiments validate the effectiveness of OpenACL and show the benefit of learning from open-world data.

**Strengths:**

1. The idea of establish novel prototype from OOD under the semi-supervised pattern is new, it involved the discovery of novel class and also utilize the provided labeled data for seen classes;

2. The experimental setting and result seems well, the result with ablation study, relative discussion make the conclusion reasonable

**Weaknesses:**

1. The recognition of the OOD/novel class when constructing the novel prototype is not clear, which make the learning process a little confuse. The author mention "the novel prototypes are used to cluster representations from novel classes", but under the setting from this paper, that OOD/Novel class may existing with labeled classes data in same task, the author applied the self-supervised learning at first to cluster each class (include OOD) [1], or recognize the novel class after see the labeled classes data [2]?

2. During the prototype-adaptation, the author remain the existing prototypes as static and not subjected to updates post clustering. Here when the model parameters updated and previous prototype may also drift [3], how should the author solve this issue?

3. The idea of applying prototype for incremental learning with open-set recognition is not novelty, especially using the contrastive learning, like [1,2,4], the author should make further discussion and comparison with recent works.


[1]. Automatically Discovering and Learning New Visual Categories with Ranking Statistics
[2]. Few-sample and adversarial representation learning for continual stream mining
[3]. Semantic Drift Compensation for Class-Incremental Learning
[4]. P-ODN: Prototype-based Open Deep Network for Open Set Recognition

**Questions:**

Please see weakness

---

> ### Author Response · Authors · 2023-11-18
> **To Review 9HZN**
>
> We thank the reviewer for the valuable comments and for acknowledging the novelty of our work. We address the reviewer's concerns below
>
> ### RE: the author applied the self-supervised learning at first to cluster each class (include OOD) [1], or recognize the novel class after see the labeled classes data [2]?
>
> In each task, there exists labeled classes and OOD/Novel classes. So, we simultaneously have seen prototypes for these labeled classes and the novel prototypes for OOD/Novel classes. OpenACL differs from both [1] and [2] in the sense that we don't separate the training for labeled class data and recognize the novel class. Instead, OpenACL handles classification and discovery at the same time, i.e., it receives both labeled data and unlabeled data in an iteration and is trained from the labeled data (by Eq.(2)) and unlabeled data (by Eq.(5)) **simultaneously**. Particularly, Eq.(5) encourages data with similar representation close to the same prototypes, so data from a novel class would approach the same novel prototype. This is a major difference from novel class discovery where most of the studies consider training on labeled data (or self-supervised training on data) and then recognize the novel class in another unlabeled dataset.
>
> ### RE: When the model parameters updated and previous prototype may also drift [3], how should the author solve this issue?
>
> We keep updating previous prototypes during the whole continual learning process. To achieve that, (1) we use replay memory to store the data of previous tasks to update previous prototypes; (2) we exploit unlabeled $\mathcal{D}_u$ which consists of data for **previous tasks**, current task, and future tasks. Therefore, our approach ensures that our previous prototypes are dynamic, preventing drift over time.
>
> Both components update the previous prototypes and help mitigate the drift. In ablation study A.2.2 (Table 5), we showed that by using the unlabeled data, OpenACL has better performance than the OpenACL(S) without using unlabeled data (e.g., it brings 6.2\% improvement in BWT). Even without using unlabeled data, OpenACL(S) still has positive BWT due to replay memory, which suggests that the previous prototypes also get updated and have better performance.

---

> > ### Author Response · Authors · 2023-11-18
> >
> > ### RE: The idea of applying prototype for incremental learning with open-set recognition is not novelty, especially using the contrastive learning, like [1,2,4], the author should make further discussion and comparison with recent works.
> >
> > Our main novelty is that we believe OOD classes in unlabeled data might become task classes in future tasks, so OOD data should be explored to improve adaptability to future tasks. Moreover, seen classes in unlabeled data can be used to tackle catastrophic forgetting. This is different from the existing works the reviewer mentioned. Specifically, [1,2] consider the standard novel class discovery problem without the open-set setting. Since they assume the unlabeled dataset only contains new classes, their proposed method have separate training phases, i.e., it first trains on a given labeled dataset and then uses the well-trained model to discover new classes. [4] uses the prototypes to detect OOD data and select high-quality ID data for the standard open-set problem. However, the discovery of new classes needs humans to identify.
> >
> > Note that none of these works applies prototypes for incremental learning with open-set recognition. To the best of our knowledge, we are the first to study continual learning in open-world settings where unlabeled data consists of past task data, current task data, future task data, and OOD data. Our goal is to exploit the extensive unlabeled data. Compared with [1,2,4], we study a more general problem including both of their settings (open-set and novel class discovery), and we don't need to introduce extra training phases or pre-train the model using the labeled dataset.  We plan to compare with [2] when the code implementation become accessible. Here we compare our algorithm with AutoNovel [1] and another contrastive learning method DVC [A] on two benchmarks as same as Table 1 and 2 in the manuscript. We adapt their methods to the Open SSCL setting and report average accuracy over three runs below:
> >
> > | Method/Task-IL    | **CIFAR-10 20%** | **CIFAR-10 50%** | **CIFAR-100 20%** | **CIFAR-100 50%** | **Tiny-ImageNet 20%** | **Tiny-ImageNet 50%** |
> > | ----------------- | ---------------------- | ---------------------- | ----------------------- | ----------------------- | --------------------------- | --------------------------- |
> > | AutoNovel         | 56.3±1.82             | 56.5±2.11             | 58.7±0.13              | 63.3±0.83              | 37.4±0.74                  | 43.1±4.74                  |
> > | DVC               | 57.4±0.86             | 61.7±3.23             | 57.6±0.92              | 62.7±2.08              | 36.8±0.61                  | 43.5±0.35                  |
> > | **OpenACL** | **64.3±2.75**   | **66.3±1.17**   | **60.4±1.19**    | **66.6±0.28**    | **40.2±0.45**        | **47.0±0.42**        |
> >
> > | **Method/Class-IL** | **CIFAR-100 20%** | **CIFAR-100 50%** | **Tiny-ImageNet 20%** | **Tiny-ImageNet 50%** |
> > | ------------------------- | ----------------------- | ----------------------- | --------------------------- | --------------------------- |
> > | AutoNovel                 | 13.2±0.61              | 17.9±1.19              | 6.5±0.57                   | 9.2±0.58                   |
> > | DVC                       | 11.2±0.78              | 16.2±2.08              | 5.8±0.40                   | 8.3±1.42                   |
> > | **OpenACL**         | **15.7±0.44**    | **20.0±1.23**    | **7.9±0.37**         | **11.9±1.06**        |
> >
> > The results further demonstrate the strong performance of OpenACL. We also compare OpenACL with other novel class discovery methods which can be found in Appendix A.4 of the latest version.
> >
> > [A] Gu, Yanan, Xu Yang, Kun Wei, and Cheng Deng. "Not just selection, but exploration: Online class-incremental continual learning via dual view consistency." In Proceedings of the IEEE/CVF Conference on Computer Vision and Pattern Recognition, pp. 7442-7451. 2022.

---

> > > ### Author Response · Authors · 2023-11-23
> > >
> > > Thank you very much for your valuable time and efforts in reviewing our paper. It is a kind reminder that **this is the last day of the Reviewer-author discussion**. Following your suggestions, we have made further discussion and comparison with recent works. Please let us know if we have addressed your concerns or if you have any additional comments. We appreciate your suggestions in this process and look forward to your reply. Thank you once again!

---

### Author Response · Authors · 2023-11-21
**To reviewers**

Thank you to each reviewer for your insightful comments. We hope our responses and updated manuscript have adequately addressed your concerns. As a gentle reminder, we are nearing the end of the discussion phase. We would greatly appreciate it if you could review our response and share your valuable feedback.

---

### Meta-Review · Area_Chair_LDYM · 2023-12-07

**Metareview:**

(a) Summarize the scientific claims and findings of the paper based on your own reading and characterizations from the reviewers.
- The paper proposes the problem of open-world semi-supervised continual learning (Open SSCL). In short, the learner obtains both labelled data from the current tasks and (IID) unlabelled data from all tasks (including future ones). Unlabelled data from future classes can be leveraged to learn their representations early.
- The authors propose OpenACL, a method for this problem that learns representations for labelled and unlabelled data using a mix of contrastive objectives.
- The authors find that the method dow comparatively better than others across three datasets in both task-IL and class-IL.

(b) What are the strengths of the paper?
- The problem of semi-supervised CL is well-motivated
- The proposed approach (OpenACL) seems reasonable and can find representation of semi-supervised data, including OOD classes
- The approach performs well, especially in the more challenging class-IL setting.

(c) What are the weaknesses of the paper? What might be missing in the submission?
- The practicality of the proposed approach remains unclear since the distribution of the unlabelled data is iid and does not change over time, and the number of classes seems like it must be known ahead of time. Showing that the representation in OpenACL works even when classes don't have full support in the unlabelled data would be useful. In the discussion, the authors also suggest that the number of prototypes does not need to equal the number of classes, but I did not find an empirical validation of the claim.
- The empirical results seem to show that the improvements might not be statistically significant in the case of Task-IL. This important nuance is not discussed in the paper, and the authors conclude that OpenACL outperforms all baselines in all settings.

**Justification For Why Not Higher Score:**

The overall assessment of the reviewers is that this is a very borderline paper.  This is reflected in the scores (average. score of 5.5) and the reviewer comments.

I find the two elements listed above under weaknesses to be rather significant, so I cannot recommend accepting the current manuscript.

**Justification For Why Not Lower Score:**

N/A

---

### Decision · Program_Chairs · 2024-01-16

Reject